# Sparser, Better, Deeper, Stronger: Improving Sparse Training with Exact Orthogonal Initialization

## Abstract

Sparse training aims to train sparse models from scratch, achieving remarkable results in recent years. A key design choice in sparse training is sparse initialization, which determines the trainable sub-network through a binary mask. Existing methods mainly revolve around selecting the mask based on predefined dense weight initialization. However, such an approach may not efficiently leverage the mask's potential impact on training parameters and optimization. An alternative direction, inspired by research into dynamical isometry, is to introduce orthogonality in the sparse subnetwork. This helps prevent the gradient signal from vanishing or exploding, ultimately enhancing the reliability of the backpropagation process. In this work, we propose Exact Orthogonal Initialization (EOI), a novel sparse orthogonal initialization scheme based on composing random Givens rotations. Contrary to other existing approaches, our method provides exact (not approximated) orthogonality and enables the creation of layers with arbitrary densities. Through experiments on contemporary network architectures, we present the effectiveness of EOI and demonstrate that it consistently outperforms other commonly used sparse initialization techniques. Furthermore, to showcase the full potential of our method, we show that it enables the training of highly sparse 1000-layer MLP and CNN networks without any residual connections or normalization techniques. Our research highlights the importance of weight initialization in sparse training, underscoring the vital part it plays alongside the sparse mask selection.

## 1 Introduction

Neural Network's compression techniques have gained increased interest in recent years in light of the development of even larger deep learning models consisting of enormous numbers of parameters. One popular solution to decrease the size of the model is to remove a portion of the parameters of the network based on some predefined criterion, a procedure known as pruning. Classical network pruning is typically performed after the training and is often followed by a fine-tuning phase (LeCun et al., 1989; Han et al., 2015; Molchanov et al., 2016)). Although such an approach decreases the memory footprint during the inference, it still requires training the dense model in the first place.

Recently, Frankle & Carbin (2019) demonstrated the existence of *lottery tickets* — sparse subnetworks that are able to recover the performance of the dense model when trained from scratch. Subsequent works have been devoted to understanding the properties of such initializations (Zhou et al., 2019; Malach et al., 2020; Frankle et al., 2020a; Evci et al., 2022) and proposing solutions that allow to obtain the sparse masks without the need to train the dense model. This gave birth to the field of sparse training (ST), which focuses on studying sparse deep neural network architectures that are already pruned at initialization – see e.g. Wang et al. (2021) for a survey. Such *sparse initializations* can be obtained either through weight-agnostic methods (e.g. random pruning (Liu et al., 2022), Erdős-Rényi initialization (Mocanu et al., 2018)), or through the use of a predefined criterion dependent on the training data or the model parameters (Lee et al., 2019b; Wang et al., 2020a; Tanaka et al., 2020).

Surprisingly, methods that apply pruning before training without any knowledge about the trained dense model can already lead to good performance (Lee et al., 2019b; Evci et al., 2020). Furthermore, even randomly removing a significant portion of the parameters at the initialization does not hurt the optimization process (Liu et al., 2022; Malach et al., 2020). Most interestingly, Frankle et al. (2020b),

alongside Su et al. (2020), demonstrated that the pruning at initialization techniques are insensitive to a random reshuffling or reinitialization of the unpruned weights within each layer. However, at the same time, Frankle et al. (2020b) also show that those techniques achieve lower accuracy than the sparse networks obtained by pruning after the full dense training. All these results motivate the research into understanding what properties make a sparse initialization successful.

In the case of dense deep neural networks models, the topic of identifying a successful initialization technique has been an intensive area of research (see for instance, Narkhede et al. (2022) for a review). One direction in this domain is given by analyzing models that attain *dynamical isometry* - have all the singular values of the input-output Jacobian close to one (Saxe et al., 2013). For some of the activation functions, this property can be induced by assuring orthogonal initialization of the weights. Networks fulfilling this condition benefit from stable gradient signal and are trainable up to thousands of layers without the use of residual connections or normalization layers (Pennington et al., 2017). Interestingly, Lee et al. (2019a) demonstrated that inducing orthogonality can also stabilize the optimization process in sparse training, a result also confirmed by Esguerra et al. (2023). However, these approaches either use only approximated isometry or are restricted in terms of compatible architectures and per-layer sparsity levels (Esguerra et al., 2023).

In this work, we propose a new orthogonal sparse initialization scheme called **E**xact **O**rthogonal **I**nitialization (EOI) that joins the benefits of exact (not approximated) isometry with the full flexibility of obtaining any sparsity level. Inspired by the work in maximally sparse matrices (Cheon et al., 2003) and sparse matrix decomposition (George & Liu, 1987; Frerix & Bruna, 2019), our method constructs the initial weights by iteratively multiplying random Givens rotation matrices. As a consequence, we sample both the mask and the initial weights at the same time. Furthermore, multiplication by an $n \times n$ Givens matrix can be performed in $O(n)$ time, which keeps the sampling process efficient. In particular, our contributions are:

- We introduce EOI (**E**xact **O**rthogonal **I**nitialization), a flexible data-agnostic sparse initialization scheme based on an efficient method of sampling random sparse orthogonal matrices by composing Givens rotations. Our method provides *exact* (not approximated) orthogonality and works both for fully-connected and convolutional layers.
- We verify that networks initialized with our method can benefit from *dynamical isometry*. In particular, we are able to train highly sparse vanilla 1000-layer MLPs and CNNs.
- We validate that imposing orthogonality in static sparse training by the use of EOI noticeably improves performance of many contemporary networks, outperforming standard pruning at initialization approaches and approximate isometry.

Our results suggest that introducing orthogonality in sparse initializations practically always improves the performance or stability of the learning, even if the underlying network already benefits from signal regularization techniques such as residual connections or normalization layers. As a consequence, we hope that our research will raise the community's awareness of the importance of simultaneously studying both the mask and weight initialization in the domain of sparse training.

## 2 RELATED WORK

**Orthogonal Initialization and Dynamic Isometry** The impact of the initialization procedure on the neural network training dynamics and performance is a prominent area of research in deep learning (Glorot & Bengio, 2010; Sutskever et al., 2013; Daniely et al., 2016; Poole et al., 2016; Narkhede et al., 2022). One studied direction in this topic is the orthogonal initialization. In particular, Saxe et al. (2013) derived exact solutions for the dynamics of deep linear networks, showing the importance of requiring the singular values of the input-output Jacobian to be equal to one, a property known as *dynamical isometry*. Those results have been later extended to nonlinear networks with tanh activations by Pennington et al. (2017), using the knowledge from the mean field theory (Poole et al., 2016; Mei et al., 2019; Schoenholz et al., 2016) and from the free random variables calculus (Voiculescu, 1995). In addition, Xiao et al. (2018) and Chen et al. (2018) studied the dynamical isometry property in CNNs and Recurrent Neural Networks, while Tarnowski et al. (2019) showed that for ResNets, dynamical isometry can be achieved naturally irrespective of the activation function. However, without skip connections or parameter sharing solutions, dynamical isometry cannot be achieved in networks with ReLU activations (Burkholz & Dubatovka, 2019). The practical speedups coming from the use of orthogonal initialization have been also proved in the work of Hu et al. (2020). Moreover, the effectiveness of orthogonal initialization has led to

many constructions developing orthogonal convolutional layers (Xiao et al., 2018; Li et al., 2019; Wang et al., 2020b; Singla & Feizi, 2021; Trockman & Kolter, 2021; Yu et al., 2021). In addition, previous studies examined signal propagation in dense or densely structured subnetworks with various regularization methods Larsson et al. (2016); Sun et al. (2022); Shulgin & Richtárik (2023). The impact of optimizing sparse weight subsets was explored by Thakur et al. Thakur et al. (2022).

**Static Sparse Training** The field of static sparse training focuses on pruning the networks at initialization. One type of methods used in this area are randomly pruned initializations, which exhibit surprisingly good performance (Liu et al., 2022; Pensia et al., 2020). Another widely used approach involves integrating weight-based knowledge with data-dependent information, like gradients, to formulate scoring functions for pruning less crucial weights. Examples include SNIP (Lee et al., 2019b), which approximates the pre and post-pruning loss, GraSP (Wang et al., 2020a) for maintaining gradient flow; and Synflow (Tanaka et al., 2020), which prevents layer collapse by using path norm principles (Neyshabur et al., 2015). In the light of this rapid development, several works aim at understanding the factors of effective sparse initializations (Frankle et al., 2020b; Su et al., 2020). In particular, Lee et al. (2019a) investigate the sparse training from the signal propagation perspective. However, their proposed method is only able to approximate orthogonality. In contrast, the recent work by Esguerra et al. (2023) demonstrates an exact-orthogonal scheme for sparse initializations, but it offers only discrete sparsity levels and isn't directly applicable to modern architectures. In comparison, our EOI initialization provides exact orthogonality and offers full flexibility in achieving various sparsity levels, combining improved performance with ease of use.

## 3 PRELIMINARIES

### 3.1 DYNAMICAL ISOMETRY

Consider a fully connected network with $L$ layers with base parameters $\mathbf{W}^l \in \mathbb{R}^{n \times n}$, where $l = 1, ..., L$. Let $\mathbf{x}^0$ denote the input to the network. The dynamics of the model can be described as: $\mathbf{x}^l = \phi(\mathbf{W}^l \mathbf{x}^{l-1} + \mathbf{b}^l)$, where $\phi$ denotes the activation function. As a consequence, the input-output Jacobian of the network is given by (Pennington et al., 2017):

$$\mathbf{J} = \frac{\partial \mathbf{x}^L}{\partial \mathbf{x}^0} = \prod_{l=1}^{L} \mathbf{D}^l \mathbf{W}^l, \text{where } \mathbf{D}^l_{i,j} = \phi'(\mathbf{h}^l_i)\delta_{i,j} \text{ and } \mathbf{h}^l = \mathbf{W}^l \mathbf{x}^{l-1} + \mathbf{b}^l \tag{1}$$

The network is said to attain *dynamical isometry* if *all* the singular values of $\mathbf{J}$ are close to one for any input chosen from the original data distribution. For linear networks, this condition is naturally fulfilled when each $\mathbf{W}^l$ is orthogonal (Saxe et al., 2013).

In the case of nonlinear networks, consider a scaled orthogonal initialization defined as $(\mathbf{W}^l)^T \mathbf{W}^l = \sigma_w^2 \mathbf{I}_n$, and $\mathbf{b}^l \sim \mathcal{N}(0, \sigma_b^2)$, where $\mathbf{I}_n$ denotes the identity matrix of size $n$. According to the mean field theory, in the large $n$ limit the empirical distribution of the preactivation converges to a Gaussian with zero mean and variance that approaches a fixed point $q_{fix}$ (Schoenholz et al., 2016). Let $\mathcal{X}$ denote the mean of the distribution of the squared singular values of matrices $\mathbf{DW}$, assuming the preactivation distribution reaches the fixed point $q_{fix}$. The network is said to be *critically initialized* if $\mathcal{X}(\sigma_w, \sigma_b) = 1$, which results in gradients that neither vanish nor explode. For tangent activations the equation $\mathcal{X}(\sigma_w, \sigma_b) = 1$ defines a line in the $(\sigma_w, \sigma_b)$ space. Moving along this line, dependable on the depth $L$, allows one to restrict the spectrum of singular values of $\mathbf{J}$, effectively achieving dynamical isometry (Pennington et al., 2017). Although the above-discussed research focused on dense models in infinite depth and width limit, recent studies have shown that influencing orthogonality also in sparse networks can lead to improved performance (Lee et al., 2019a; Esguerra et al., 2023).

### 3.2 ORTHOGONAL CONVOLUTIONS

Analogously to standard linear orthogonal layers, in convolutional layers, orthogonality is defined by ensuring that $||\mathbf{W} * \mathbf{x}||_2 = ||\mathbf{x}||_2$ holds for any input tensor $\mathbf{x}$, where $\mathbf{W} \in \mathbb{R}^{c_{out} \times c_{in} \times (2k+1) \times (2k+1)}$ represents the convolutional weights, and $*$ denotes the convolution operation (Xiao et al., 2018). Some examples of initializations fulfilling this condition are the *delta-orthogonal* initialization (Xiao et al., 2018), the Block Convolutional Orthogonal (BCOP) initialization (Li et al., 2019), or the

Explicitly Constructed Orthogonal (ECO) convolutions (Yu et al., 2021). In this work, we focus on the use of the *delta-orthogonal* method, due to its pioneering impact on the study of dynamical isometry in CNNs, as well as a relatively simple and intuitive construction. The *delta-orthogonal* approach involves embedding an orthogonal matrix $\mathbf{H}$ into the central elements of the convolutional layer's kernels, while setting all other entries to zero. Formally, the convolution weights $\mathbf{W}$ can be expressed as $\mathbf{W}_{i,j,k,k} = \mathbf{H}_{i,j}$ ,where $\mathbf{H}$ is a randomly selected dense orthogonal matrix with shape $c_{out} \times c_{in}$.

Let us also note that the standard convolution operation is in practice implemented by multiplying the kernel $\mathbf{W}$ with the *im2col* representation of the data (Heide et al., 2015; Pytorch-Conv2d). Some methods explored solutions that penalize the distance $||\mathbf{W}^T\mathbf{W} - \mathbf{I}||_2$, stabilizing the Jacobian of such multiplication (Xie et al., 2017; Balestriero & Baraniuk, 2018). However, this approach doesn't necessarily guarantee orthogonality in terms of norm preservation, as it does not account for the change of the spectrum introduced by applying the *im2col* transformation (Wang et al., 2020b).

### 3.3 STATIC SPARSE TRAINING

In static sparse training, the network is pruned during the *initialization* phase and its structure is kept intact throughout the training process. Let $M$ denote the binary mask with zero-entries representing the removed parameters. The density $d$ of the model is defined as $d = ||M||_0/m$, where $|| \cdot ||_0$ is the $L_0$-norm and $m$ is the total number of parameters in the network. As a result, the sparsity $s$ of the model can be calculated as $s = 1 - d$. Different static sparse training methods vary in how they select per-layer masks $\mathbf{M}^l$ and, consequently, per-layer densities $d_l$. Note that we require that $d = \sum_l^L (d_l m_l)/m$, where $m_l$ denotes the total number of parameters in layer $l$. Below, we summarize some common sparse training approaches studied within this work:

**Uniform** - In this simple method each layer is pruned to density $d$ by randomly removing $(1 - d)n_l$ parameters, where $n_l$ denotes the $l$-th layer's size.

**Erdős-Rényi (ERK)** - This approach randomly generates the sparse masks so that the density in each layer $d^l$ scales as $\frac{n^{l-1}+n^l}{n^{l-1}n^l}$ for a fully-connected layer (Mocanu et al., 2018) and as $\frac{n^{l-1}+n^l+w^l+h^l}{n^{l-1}n^l w^l h^l}$ for a convolution with kernel of width $w^l$ and height $h^l$ (Evci et al., 2020).

**SNIP** - In this method each parameter $\theta$ is assigned a score $\rho(\theta) = |\theta\nabla_\theta\mathcal{L}(D)|$, where $\mathcal{L}$ is the loss function used to train the network on some data $D$. Next, a fraction of $s$ parameters with the lowest score is removed from the network (Lee et al., 2019b).

**GraSP** - The Gradient Signal Preservation (GraSP) score is given by $\rho(-\theta) = -\theta \odot \mathbf{G}\nabla_\theta\mathcal{L}(D)$, where $\mathbf{G}$ is the Hessian matrix, and $\odot$ denotes the element-wise multiplication. Next, the top $s$ parameters with the highest scores are removed from the network (Wang et al., 2020a).

**Synflow** - This approach proposes an iterative procedure, where the weights are scored using $\rho(\theta) = \nabla_\theta R_{SF} \odot \theta$. The term $R_{SF}$ is a loss function expressed as $R_{SF} = \mathbb{1}^T \prod_{l=1}^L |\theta^l| \mathbb{1}$, where $\mathbb{1}$ is the all-ones vector, $|\theta^l|$ is the element-wise absolute value of all the parameters in layer $l$, and $L$ is the total number of layers (Tanaka et al., 2020).

Note that every static sparse training method automatically defines also a *sparse initialization*, which is the element-wise multiplication of the weights with their corresponding masks. It has been shown that the pruning at initialization methods are invariant to the re-initialization of the parameters or the reshuffling of the masks within a layer (Frankle et al., 2020b; Su et al., 2020). Consequently, one can treat them as a source of the per-layer densities $d_1, \ldots, d_L$, inferred from their produced masks. Throughout this work we will therefore refer to the aforementioned methods as *density distribution* algorithms. Finally, our goal is to also compare the proposed EOI scheme with other sparse initialization schemes. In particular, we investigate:

**Approximated Isometry** - Given the per layer masks $\mathbf{M}^l$ form a density distribution algorithm, the Approximated Isometry (AI) scheme optimizes the weights in the masks by minimizing the orthogonality loss $||(\mathbf{W}^l \odot \mathbf{M}^l)^T(\mathbf{W}^l \odot \mathbf{M}^l) - \mathbf{I}||_2$ with respect to layer weights $\mathbf{W}^l$. Although simple in design and implementation, such an approach may suffer from slow optimization procedure and approximation errors. Furthermore, as discussed in section 3.2, the used by AI orthogonality loss is ill-specified in terms of convolutions.

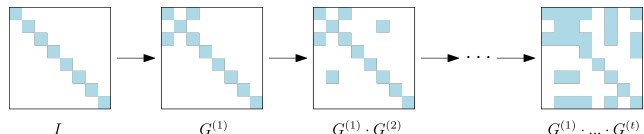

Figure 1: Sparse orthogonal matrix generation via composition of Givens rotations.

**SAO** - The Sparsity-Aware Orthogonal initialization (SAO) scheme constructs a sparse matrix by first defining the non-zero mask structure via a bipartite Ramanujan graph. Next, the obtained structural mask is assigned orthogonal weights, to assure isometry. In the case of convolutional layers, SAO performs structural-like pruning, by sampling the middle orthogonal matrix of the delta orthogonal initialization and removing filters with only zero entries. The drawback of SAO is that it constrains the possible dimensionality of the weights and inputs. Furthermore, it supports only discrete increases in the sparsity levels. Consequently, this scheme is hard to adapt to contemporary architectures. See Appendix H for detailed description of SAO.

## 4 EXACT ORTHOGONAL INITIALIZATION

In this work, we present a novel sparse orthogonal initialization scheme, called Exact Orthogonal Initialization (EOI) compatible with both fully-connected and convolutional layers. Our solution combines the advantages of ensuring exact isometry with complete flexibility in achieving arbitrary sparsity levels. This is accomplished by generating the sparse orthogonal matrices through a composition of random Givens rotations. We start this section by explaining this construction and follow with a discussion of its use in static sparse training.

### 4.1 RANDOM, SPARSE, ORTHOGONAL MATRICES

A Givens rotation is a linear transformation that rotates an input in the plane spanned by two coordinate axes. The Givens matrix of size $n$ for indices $1 \leq i < j \leq n$ and angle $0 \leq \varphi < 2\pi$, denoted as $G_n(i, j, \varphi)$, has a structure similar to identity matrix of size $n$, with only four entries overwritten: $G_n(i, j, \varphi)_{ii} = G_n(i, j, \varphi)_{jj} = \cos \varphi$, $G_n(i, j, \varphi)_{ij} = -\sin \varphi$, and $G_n(i, j, \varphi)_{ji} = \sin \varphi$. Givens matrices can be randomly sampled by uniformly selecting a pair $(i, j)$ where $i < j$, along with an angle $\varphi$. Importantly, the multiplication of such a rotation with any matrix of size $n$ can be computed in $O(n)$ time. This property has made Givens matrices a popular choice in implementing linear algebra algorithms, such as, for instance, the QR decomposition.

Since Givens matrices are orthogonal, their product is also orthogonal. Moreover, iteratively composing random Givens rotations produces denser outputs. Due to those facts, Givens matrices are commonly used in the study of (random) sparse orthogonal matrices (Cheon et al., 2003; George & Liu, 1987; sprand MatLab). In particular, by leveraging the above observations, we can sample an orthogonal matrix by initiating our process with an identity matrix $A \leftarrow I_n$, which represents the currently stored result. Then, we randomly select a Givens rotation $G_n(i, j, \varphi)$ and multiply it with the current result, substituting $A \leftarrow A \cdot G_n(i, j, \varphi)$. We repeat this operation until we achieve the desired target density $d$ of the matrix. We outline this approach in Algorithm 1 and visualize the premise behind it in Figure 1.

The running time of the proposed method depends on the number of passes we need to perform in line 2 of the Algorithm. This means that we are interested in how quickly the density of the resulting matrix $A$ increases with the number of random Givens multiplications $t$. We observe that:

**Observation 1.** *Let $A^{(t)}$ be a random $n \times n$ matrix representing the product of $t$ random, independent Givens matrices of size $n$. Then the expected density $\mathbb{E}[\mathrm{dens}(A^{(t)})]$ of matrix $A^{(t)}$ is given by:*

$$\mathbb{E}[\mathrm{dens}(A^{(t)})] = \frac{1}{n} \cdot \sum_{k=1}^{n} k \cdot p(t, k), \tag{2}$$

*where $p(t, k)$ denotes the probability that a fixed row of $A^{(t)}$ will have $k$ non-zero elements. The values $p(t, k)$ are independent of the choice of the row, and are defined by the recurrent relation:*

$$p(t + 1, k + 1) = p(t, k + 1) \cdot \frac{\binom{k+1}{2} + \binom{n-k-1}{2}}{\binom{n}{2}} + p(t, k) \cdot \frac{k \cdot (n - k)}{\binom{n}{2}} \tag{3}$$

*with the base condition $p(0, 1) = 1$ and $p(0, k) = 0$ for $k \neq 1$.*

For proof please refer to Appendix C. Since the above result is not immediately intuitive, we illustrate the expected density computed via Equation 2 for a matrix of size $100 \times 100$ in Figure 2. Our mathematical derivations perfectly match the empirical observations. The density smoothly rises with the number of used rotations, being initially convex up to a critical point (here it is around $t \approx 270$), and then becoming concave, which indicates a sigmoidal relation. Therefore achieving high densities would necessitate a substantial number of iterations. However, in sparse training, our focus typically lies within highly-sparse regions, which often do not exceed a density of 50%. This keeps the algorithm very efficient.

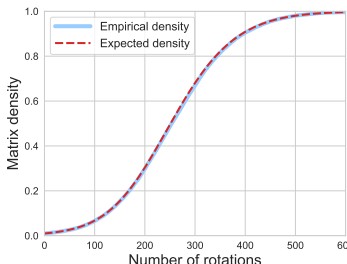

Figure 2: Expected density (red) of a sparse matrix produced by Algorithm 1 as as a function of the number of applied Givens rotations. Blue curve represents the empirical evaluation.

---

**Algorithm 1** Generate a random $n \times n$ orthogonal matrix of target density $d$.

---

**Require:** $n \geq 2, d \in [0, 1]$
1:   $A \leftarrow I_n$      $\triangleright$ identity matrix of size $n$
2:   **while** $\mathrm{dens}(A) < d$ **do**
3:      Pick $(i, j)$ such that $1 \leq i < j \leq n$,
4:         uniformly, at random.
5:      Pick $0 \leq \varphi < 2\pi$,
6:         uniformly, at random.
7:      $A \leftarrow A \cdot G_n(i, j, \varphi)$
8:   **end while**
9:   **return** $A$

---

## 4.2 EXACT ORTHOGONAL INITIALIZATION FOR STATIC SPARSE TRAINING

Consider a neural network with $L$ layers, with associated per-layer densities $d^1, \ldots, d^L$ given by some density distribution algorithm. The Exact Orthogonal Initialization scheme processes each layer independently. If layer $l$ is fully connected, we proceed by sampling an orthogonal matrix $\mathbf{W}^l$ with the target density $d^l$ using Algorithm 1. In the case of a non-square weight matrix with size $n \times m$, where $n < m$, we sample a row-orthogonal matrix by appending additional $m - n$ columns of zeros to the initial $A$ and using Givens rotations of size $m$. If $n > m$ we use the same procedure to obtain a $m \times n$ row-orthogonal matrix and transpose the output. We treat all zeros of the matrix as pruned connections so that $\mathbf{M}_{ij}^l = 0$ if and only if $\mathbf{W}_{ij}^l = 0$.

In the case of a convolutional layer of shape $c_{out} \times c_{in} \times (2k + 1) \times (2k + 1)$, we sample a sparse orthogonal matrix $\mathbf{H}^l$ of density $d^l$ with shape $c_{out} \times c_{in}$ using the same procedure as described above. Next, we adapt the delta orthogonal initialization and set $\mathbf{W}_{i,j,k,k}^l = \mathbf{H}_{i,j}^l$ and $\mathbf{W}_{i,j,p,q}^l = 0$ if any of $p, q$ is different than $k$. To create the sparse mask $\mathbf{M}^l$, we follow a two-step process. First, we set $\mathbf{M}_{i,j,p,q}^l = [\mathbf{W}_{i,j,p,q}^l \neq 0]$. However, the resulting mask from this step has a density of $d^l/(2k + 1)^2$ which is too low, and the only learnable parameters are located in the centers of the filters within $\mathbf{W}^l$. To rectify this, in the next step, we select uniformly, at random a sufficient number of zero entries in $\mathbf{M}^l$ and change them to 1. This adjustment ensures that the desired density is achieved while also spreading the learnable parameters of $\mathbf{W}^l$ throughout the tensor. It's worth noting that this approach also allows us to sample matrices $\mathbf{H}^l$ with densities higher than the target $d^l$, such as $d_{\mathbf{H}^l} = \sqrt{d^l}$, by compensating for the change with a higher level of sparsity in the non-central entries of the mask.

Finally, in both the fully connected and convolutional cases, we scale the obtained weights by $\sigma_w$, following the practice from dense orthogonal initializations. The bias terms are sampled from a normal distribution with zero-mean and standard deviation $\sigma_b$. See Appendix F for a detailed description and visualization of the method.

## 5 EXPERIMENTS

In this section we empirically assess the performance of our method. We start by analyzing the properties of the input-output Jacobian to verify the improvement in signal propagation statistics. Next, we showcase the potential of our initialization scheme by training highly sparse vanilla 1000-layer networks. Finally, we demonstrate the practical benefits of out EOI scheme in sparsifying contemporary architectures. See Appendix A for details on the implementation and training regime encompassed in each conducted study.

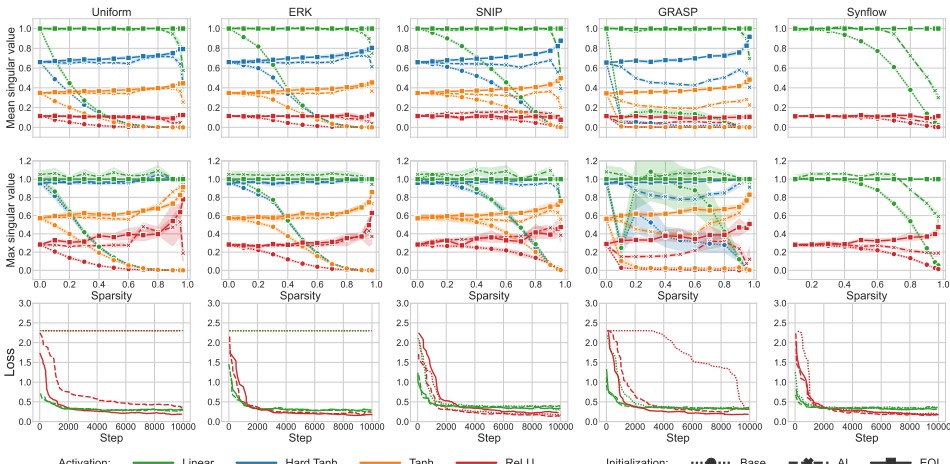

Figure 3: The mean (top row) and maximum (middle row) singular values of the input-output Jacobian of an MLP network computed for varying sparsity levels. In addition, we also present the training loss curve (bottom row) for sparsity 0.95. The colors indicate the used activation function, while the line- and marker-styles represent the initialization schemes. In the loss curve plots, for clarity of the presentation, we show only the ReLU and linear activation. See Appendix D for other activations.

## 5.1 STUDY OF SINGULAR VALUES

To evaluate our method in the context of dynamical isometry theory, we examine the singular values of the input-output Jacobian of an MLP network with 7 layers and a hidden size of 100. We use this architecture as it was also studied for the same purposes in the work of Lee et al. (2019a). We start by initializing the dense model to have orthogonal weights. Next, we run the density distribution methods described in Section 3.3 to produce the sparse masks. We compare the sparse initializations obtained by directly applying these masks (Base) with the initializations returned by the approximated isometry scheme (AI) and our Exact Orthogonal Initialization scheme (EOI). We report the mean and maximum singular values for a set of various sparsity levels, ranging from 0% (dense network) up to 97%. We explore different activation functions, including linear (no activation), tanh, hard tanh, and ReLU. Additionally, we also investigate the training loss for a sparsity level of 95% in order to validate the impact of the orthogonality on the dynamics of learning. We present the results in Figure 3.

For the Base initialization, the mean and maximum singular values decrease with the increase of the sparsity, irrespective of the used activation function, as was also witnessed in Lee et al. (2019a). Interestingly, we find that some density distribution methods are more resilient to this behavior. In particular, the Synflow method is able to maintain good statistics of the input-output Jacobian up to a sparsity of 0.5%. Moreover, we note that imposing orthogonality, whether through AI or our EOI, aids in preserving nearly identical mean and maximum singular values compared to the dense model. However, our approach is much better at sustaining this behavior even within the exceedingly high sparsity range (beyond 0.95). We argue that this distinction arises from the exactness of our orthogonality, whereas AI is susceptible to approximation errors.

Apart from measuring the singular values we verify the learning capabilities of the studied networks (see the bottom row of Figure 3). We observe that in practically every case, sparse orthogonal initialization helps to achieve better training dynamics, especially in the initial steps of the optimization. However, we find that this effect is less eminent for the Synflow and SNIP methods. Interestingly, we noted that, in the majority of the situations, the training curve improves even in the case of the ReLU activation. In consequence, even if dynamical isometry is not possible to achieve, sparse orthogonal initializations can still enhance the training procedure of the network.

## 5.2 1000-LAYER NETWORKS

To showcase the effectiveness of our initialization scheme, we carry out experiments on extremely deep networks with the tanh activation function. We use two models: a 1000-layer MLP with a width of 128 and a 1000-layer CNN with a hidden channel size of 128. We do not use any residual

connections or normalization layers. We initialize the baseline dense models to achieve dynamical isometry. More precisely, we use the orthogonal initialization (delta-orthogonal with circular padding in case of the CNN) and set the $(\sigma_w, \sigma_b)$ parameters of that initialization to reach criticality, i.e. $\mathcal{X}(\sigma_w, \sigma_b) = 1$. To this end, we follow the procedure discussed in Xiao et al. (2018).[1]

To obtain the sparse baselines we employ the density distribution algorithms discussed in Section 3.3 for sparsity of 87.5%(column **Method** in Table 1)[2]. Apart from directly adapting the sparse initializations provided by those methods ("Base" in column **Init**), we also extract their density-per-layer ratios to produce the sparse orthogonal initializations using the AI and our EOI schemes. The chosen sparsity level of 87.5% and network sizes also enable us to incorporate the SAO method into this study. However, this method cannot be used with varying per-layer sparsities and comes with its own density distribution, which is equal to the uniform one, with the first and last layers unpruned. In addition, we also evaluate EOI on the SAO density distribution. We present the results in Table 1.

Table 1: Test accuracy of 1000-layer MLP and CNN networks on MNIST and CIFAR-10 respectively, density 12.5%. We bold-out the best result within each density distribution method and architecture, and underline the best overall result within each architecture.

| Method | Init | MLP (MNIST) | CNN (CIFAR10) |
|---|---|---|---|
| ERK | Base | 11.35(0.00) | 10.00(0.00) |
| | AI | 95.73(0.10) | 39.08(16.35) |
| | EOI | **95.98(0.25)** | **78.41(0.15)** |
| GraSP | Base | 11.35(0.00) | 16.05(0.40) |
| | AI | 95.33(0.11) | 14.48(1.12) |
| | EOI | **95.68(0.30)** | **77.68(0.34)** |
| SNIP | Base | 11.35(0.00) | 10.00(0.00) |
| | AI | **95.35(0.40)** | 10.00(0.00) |
| | EOI | 94.06(1.11) | **46.59(0.11)** |
| Uniform | Base | 11.35(0.00) | 10.00(0.00) |
| | AI | 95.52(0.30) | 33.35(21.34) |
| | EOI | **96.11(0.20)** | **76.87(0.34)** |
| SAO | Base | 96.74(0.13) | 78.80(0.11) |
| | EOI | **97.43(0.05)** | **79.60(0.24)** |
| Dense | Orthogonal | 97.51(0.13) | 80.71(0.25) |

We observe that directly using the sparse mask obtained from the density distribution algorithms ("Base" in the Table) leads to random performance. This is expected since, as observed in Section 5.1, the performed pruning causes the degradation of the signal propagation. Applying orthogonal-based sparse initialization schemes helps to recover good performance. Most notably, our EOI consistently outperforms other sparse approaches and holds the overall best sparse performance (denoted by underscore in Table) in both models. Importantly, exact orthogonalization holds a clear advantage over the approximated one. We argue that, in case of MLPs, this is due to the accumulated approximation errors. In the case of the CNN, the AI approach fails due to its ill-specification of the orthogonality of a convolutional layer. The SAO method, on the other hand, can achieve high results, however, it does not hold any clear advantage over EOI. This study showcases that, with the use of EOI, the efficacy of orthogonal initializations and dynamical isometry can be extended to sparse training.

## 5.3 PRACTICAL BENEFITS OF ORTHOGONALITY

In the previous section, we showed that our EOI initialization can excel in setups related to studies of dynamical isometry. In this section, we demonstrate that orthogonality can also improve the sparse performance in more commonly used architectures that employ residual connections and normalization techniques. To this end, we study a selection of models frequently employed in sparse studies, such as the VGG-16 (Simonyan & Zisserman, 2014), ResNet32, ResNet56, and ResNet110 (He et al., 2016) on CIFAR-10 (Krizhevsky et al., 2009), as well as EffcientNet (Tan & Le, 2019) on the Tiny ImageNet dataset (Le & Yang, 2015). For the baseline dense models, we study both their base initialization, as well as the orthogonal one. In the sparse case, similarly to the experiments from Section 5.2, we pair density distribution methods with weight / mask initializations to form sparse initialization algorithms. Note that we are not able to use SAO in this study, due to the constraints it introduces for the possible density levels and network architectures. In total we evaluate 15 variants of sparse initialization, 5 of which are EOI-based. We present the test accuracies in Table 2.

Interestingly, observe that the orthogonal approaches consistently outperform the Base one. The only exception is the Efficient Net model. We argue that this may be due to the bottleneck blocks used in this architecture, which prevent proper orthogonal initializations. At the same time, we notice that the discrepancies between the AI and EOI are much smaller in this study than in Section 5.2. This

---

[1]See Appendix A exact values of the initialization parameters

[2]When computing the scores for the density distribution algorithms for the CNN, we use the orthogonal convolution approach from Algorithm 1 of Xiao et al. (2018) – see Appendix A.

Table 2: Test accuracy of various convolutional architectures on dedicated datasets, density 10%. We bold-out the best result within each density distribution method and architecture, and underline the best overall result for each architecture-activation pair.

| Method | Init | ResNet32 (CIFAR10) | ResNet56 (CIFAR10) | ResNet110 (CIFAR10) | VGG-16 (CIFAR10) | EfficientNet (Tiny ImageNet) |
|---|---|---|---|---|---|---|
| ERK | Base | 89.49(0.27) | 90.82(0.23) | 91.51(0.32) | 91.44(0.19) | **45.54(0.76)** |
| | AI | 89.51(0.21) | 90.74(0.17) | **91.93(0.20)** | 91.51(0.10) | 45.13(0.70) |
| | EOI | **89.79(0.38)** | **91.07(0.15)** | 91.66(0.35) | **92.02(0.12)** | 44.13(0.77) |
| GraSP | Base | 88.95(0.21) | 90.18(0.35) | 91.60(0.50) | 33.22(23.11) | 46.80(0.35) |
| | AI | 89.26(0.33) | 90.66(0.35) | 91.51(0.53) | 34.87(21.97) | **46.98(0.43)** |
| | EOI | **89.66(0.09)** | **90.89(0.20)** | **91.82(0.17)** | **92.65(0.28)** | 45.26(0.29) |
| SNIP | Base | 89.49(0.27) | 90.57(0.31) | 91.44(0.31) | **92.62(0.17)** | 22.66(1.26) |
| | AI | 89.48(0.16) | 90.46(0.20) | **91.49(0.39)** | 92.55(0.20) | 22.93(1.20) |
| | EOI | **89.54(0.39)** | **90.79(0.20)** | 91.33(0.33) | 92.58(0.17) | **50.53(0.65)** |
| Synflow | Base | 88.78(0.27) | 90.05(0.23) | 91.33(0.24) | 92.15(0.19) | 53.19(0.63) |
| | AI | 88.82(0.24) | 90.31(0.29) | 91.55(0.22) | 92.33(0.16) | **52.70(0.31)** |
| | EOI | **89.49(0.38)** | **90.96(0.11)** | **91.86(0.11)** | **92.36(0.24)** | 52.69(0.56) |
| Uniform | Base | 88.27(0.27) | 89.64(0.23) | 91.11(0.38) | 90.50(0.14) | **30.93(0.19)** |
| | AI | 88.25(0.33) | 89.86(0.08) | 91.16(0.28) | 90.55(0.14) | 22.72(13.36) |
| | EOI | **89.08(0.40)** | **90.42(0.26)** | **91.42(0.43)** | **90.91(0.26)** | 24.51(13.25) |
| Dense | Base | 92.68(0.19) | 93.01(0.23) | 92.52(0.47) | 92.84(0.08) | 52.98(0.43) |
| | Orthogonal | 92.73(0.33) | 92.59(0.87) | 92.79(0.85) | 92.84(0.23) | 53.05(0.73) |

can be attributed to the fact that skip connections and normalization layers already facilitate learning stabilization. In consequence, the ill-specification of the convolution orthogonality is less detrimental in this case. Remarkably, within each density-distribution method, we observe that our EOI scheme typically yields the highest test accuracy. This is evident from the mean rank assigned after evaluating the results of each scheme across all density distribution methods and models (refer to Figure 4).

The above results advocate for the use of sparse orthogonal schemes, even if the network already has good signal propagation properties. This may suggest that influencing orthogonality in very sparse regimes plays yet another role, apart from simply stabilizing the learning.

## 6 CONCLUSIONS

In this work, we introduced the Exact Orthogonal Isometry (EOI)—a novel sparse orthogonal initialization scheme compatible with both fully-connected and convolutional layers. Our approach is the first orthogonality-based sparse initialization that, at the same time, provides exact orthogonality, works perfectly for convolutional layers, supports any density-per-layer, and does not unnecessarily constrain the layer's dimensions. As a result, it surpasses approximated isometry in high-sparsity scenarios while remaining easily adaptable to contemporary networks and compatible with any density distribution. Experimental analysis demonstrates that our method consistently outperforms other popular static sparse training approaches and enables the training of very deep vanilla networks by benefiting from dynamical isometry. We believe that the remarkable performance of EOI underscores the potential of employing orthogonal initializations in sparse training. Furthermore, it highlights the critical need to take into account not only the binary mask values but also the weight initialization characteristics that arise (or disappear) as a result of pruning.

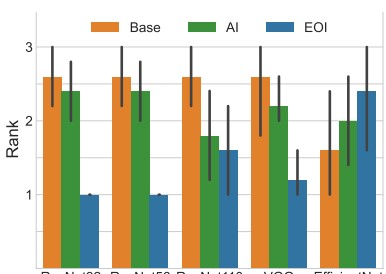

Figure 4: Ranking of initialization schemes for different models. The *lower* the value, the better. EOI performs the best on the 4 out 5 cases.

**Limitations and Future Work**. Our study focused on image recognition tasks, in order to be comparable to other works in this domain. It would be interesting to investigate the possible applicability of our sparse orthogonal schemes to large NLP models. In addition, future efforts can be directed towards manipulating the distribution of random Givens matrices or adjusting the central orthogonal matrix's density in convolutional layers for improved results. In a broader context, there's significant potential for research in sparse training, especially in developing methods that sample weights and masks simultaneously.

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

# A  TRAINING REGIME

In all the conducted experiments, we set the batch size to 128 and use the SGD optimizer, with a momentum parameter set to 0.9. We conducted each experiment independently five times to ensure the robustness and reliability of the results. We do not optimize any specific hyperparameters. In the following Subsections, we discuss the design choices specific to each experiment setup from the main text.

## A.1  STUDY OF SINGULAR VALUES

For the experiments in Section 5.1 we used a simple MLP network with 7 layers and width 100 trained on the MNIST dataset (LeCun et al., 1998). The same architecture was used in Lee et al. (2019a) for an analogous evaluation. We train the network for 100 epochs, employing a learning rate of $10^{-2}$ and a weight decay of $10^{-4}$. We set the parameters $\sigma_w$ and $\sigma_b$ from dynamical isometry to 1 and 0, respectively. For AI (approximate isometry), we conduct $10^4$ iterations of the orthogonality optimization process, which is the default value also used by Lee et al. (2019a). We test sparse models with density from the set $\{0.03, 0.05, 0.1, 0.2, 0.3, 0.4, 0.5, 0.6, 0.7, 0.8, 0.9, 1.0\}$. and report the mean and maximal singular value of the input-output Jacobian.

## A.2  1000-LAYER NETWORKS

In experiments discussed in Section 5.2 we investigated extremely deep MLPs and CNNs. In all experiments within this category, we configure the $\sigma_w$ and $\sigma_b$ parameters from dynamical isometry to be $1.0247$ and $0.00448$, respectively. In order to obtain those values we use the scheme discussed in (Schoenholz et al., 2016; Xiao et al., 2018).

The MLP architecture consists of 1000 layers, with 998 hidden layers, each having a shape of $128 \times 128$, taking inspiration from the model used in Pennington et al. (2017). We train the MLP on the MNIST dataset for 100 epochs, using a learning rate of $10^{-4}$ and weight decay of $10^{-4}$.

In the case of the 1000-layer CNN architecture, the first layer consists of an input transformation layer that increases the channel size to 128. The next two layers are convolutions with stride 2, that perform dimension reduction. The last layers consist of average pooling, flattening layer and a fully-connected layer. The remaining middle layers are always convolutions with a kernel size of 3 and input and output channel width of 128. In all convolution layers in this model, we use circular padding. This construction was inspired by the similar architecture studied in (Xiao et al., 2018).

We train the CNN model for 100 epochs using a learning rate of $10^{-4}$ and weight decay of $10^{-4}$. Additionaly, In experiments involving EOI, we sampled the central orthogonal matrices $\mathbf{H}^l$ with a density of $d_{\mathbf{H}^l} = d^l$. For experiments with AI, we set the number of orthogonality optimization iterations to $10^3$. Moreover, when computing the scores for the density distribution algorithms for the CNN, we use the orthogonal convolution approach from Algorithm 1 of Xiao et al. (2018), instead of the delta-orthogonal one. Note that both solutions adhere to the convolution definition from Section 3.2. However, the delta initialization results in more weights with zero-magnitude and hence would unnecessarily bias the score-based density distribution algorithms, such as SNIP or GraSP. For the dense baseline, we use the delta orthogonal scheme.

### A.3 RESNETS, VGG, EFFICIENTNET

In this Section, we describe the setting used in experiments presented in Section 5.3. In this study, we use the VGG-16 (Simonyan & Zisserman, 2014), ResNet-32/56/110 (He et al., 2016), as Efficient-Net-B0 (Tan & Le, 2019) architectures. Note that for the VGG model, we adapt the VGG16-C variant introduced in Dettmers & Zettlemoyer (2019). We employ the following common setups:

We train the VGG and ResNet models on the CIFAR-10 (Krizhevsky et al., 2009) dataset for 200 epochs with a learning rate of 0.1 and weight decay of $10^{-4}$. We set the $\sigma_w$ and $\sigma_b$ parameters to 1 and 0, respectively. In experiments involving EOI, we chose to sample denser orthogonal matrices $\mathbf{H}^l$ to be used as centers of convolu-

Table 3: Number of parameters of the sparsified architectures used in the experiments.

| Architecture | Density | #Params |
|---|---|---|
| ResNet-32 | 10% | $\sim 45K$ |
| ResNet-56 | 10% | $\sim 85K$ |
| ResNet-110 | 10% | $\sim 170K$ |
| VGG16-C | 10% | $\sim 1.1M$ |
| EfficientNet | 10% | $\sim 420K$ |
| 1000-layer-MLP | 12.5% | $\sim 2M$ |
| 1000-layer-CNN | 12.5% | $\sim 18M$ |

tional tensors, Specifically, we set the density $d_{\mathbf{H}^l}$ to $\sqrt{d^l}$, as we observed that it leads to improved performance compared to setting $d_{\mathbf{H}^l} = d^l$. Similarly to the setup discussed in Section A.2, we also use Algorithm 1 of Xiao et al. (2018) to initialize the convolution when computing the density distribution algorithms. Here, we also use it in the dense baseline. For experiments with AI, we configure the number of orthogonality optimization iterations to be $10^4$. Additionally, in experiments involving Synflow on ResNet-56 and ResNet-110, we use parameters of double precision.

For experiments involving the EfficientNet-B0 (Tan & Le, 2019) architecture, we used the Tiny-ImageNet (Le & Yang, 2015) dataset. The overall configuration follows the one on ResNets and VGG, with the exception that we train the network for 120 epochs using a learning rate of $10^{-2}$ and a weight decay of $10^{-3}$.

Finally, let us note that throughout our work, we consistently employ the default training and test splits provided by the datasets. However, since the default splits of MNIST and CIFAR-10 do not include a validation set, we use 10% of the training split as the validation dataset and the remaining 90% for the actual training dataset.

## B EFFICIENCY OF EOI

In addition to comparing performance, we also assess EOI in terms of the compute time and the quality of the resulting orthogonal initializations when compared to AI and SAO.

To evaluate the practical time complexity, we consider the task of generating a sparse orthogonal matrix $A$ with dimension $n \times n$. We evaluate the EOI, SAO and AI schemes for matrices with size varying from $n = 16$ to $n = 2048$. We record the actual wall-time required to obtain the sparse isometry for each method and matrix size. Additionally, for each such matrix, we report the mean Orthogonality Score (OS), which is defined as the norm of the difference between $A^T A$ and the

Table 4: Test accuracy of ResNet32 and VGG-16 on CIFAR-10, using various per-layer density distribution algorithms, density 10%. We bold-out the best result within each density distribution method and architecture, and underline the best overall result within each architecture.

| Method | Init | ResNet32 | | | | VGG-16 | | | |
|---|---|---|---|---|---|---|---|---|---|
| | | ReLU | LReLU | SELU | Tanh | ReLU | LReLU | SELU | Tanh |
| ERK | Base | 89.49(0.27) | 89.41(0.30) | 89.24(0.16) | 87.96(0.20) | 91.44(0.19) | 91.41(0.25) | 90.70(0.30) | **91.21(0.12)** |
| | AI | 89.51(0.21) | 89.36(0.19) | 89.22(0.25) | **88.30(0.18)** | 91.51(0.10) | 91.61(0.14) | 90.80(0.26) | 91.09(0.22) |
| | EOI | **89.79(0.38)** | **89.81(0.08)** | **89.37(0.14)** | 88.06(0.12) | **92.02(0.12)** | **92.21(0.25)** | **91.16(0.12)** | 91.13(0.27) |
| GraSP | Base | 88.95(0.21) | 88.92(0.38) | 89.02(0.40) | **88.18(0.32)** | 33.22(23.11) | 92.64(0.20) | 91.29(0.22) | 91.65(0.12) |
| | AI | 89.26(0.33) | 89.03(0.26) | 89.18(0.24) | 88.16(0.23) | 34.87(21.97) | **92.57(0.15)** | 91.10(0.18) | **91.79(0.16)** |
| | EOI | **89.66(0.09)** | **89.59(0.26)** | **89.39(0.25)** | 88.17(0.26) | **92.65(0.28)** | 92.48(0.29) | **91.42(0.21)** | 91.71(0.14) |
| SNIP | Base | 89.49(0.27) | 89.29(0.49) | 89.05(0.22) | 87.91(0.30) | **92.62(0.17)** | **92.80(0.20)** | 91.51(0.12) | 92.00(0.23) |
| | AI | 89.48(0.16) | 89.42(0.19) | 88.91(0.11) | 87.73(0.25) | 92.55(0.20) | 92.74(0.14) | 92.55(0.09) | 91.83(0.06) |
| | EOI | **89.54(0.39)** | **89.57(0.21)** | **89.40(0.17)** | **87.90(0.57)** | 92.58(0.17) | 92.55(0.09) | **91.74(0.10)** | **92.08(0.15)** |
| Synflow | Base | 88.78(0.27) | 88.90(0.26) | 88.56(0.19) | - | 92.15(0.19) | 92.30(0.16) | **91.84(0.20)** | - |
| | AI | 88.82(0.24) | 88.85(0.41) | 88.48(0.22) | - | 92.33(0.16) | 92.33(0.12) | 91.51(0.13) | - |
| | EOI | **89.49(0.38)** | **89.46(0.25)** | **89.13(0.21)** | - | **92.36(0.24)** | **92.63(0.24)** | 91.69(0.29) | - |
| Uniform | Base | 88.27(0.27) | 88.19(0.19) | 88.27(0.11) | 87.16(0.19) | 90.50(0.14) | 90.48(0.19) | 89.96(0.14) | 89.88(0.16) |
| | AI | 88.25(0.33) | 88.28(0.38) | 88.38(0.22) | 87.11(0.24) | 90.55(0.14) | 90.33(0.25) | 90.02(0.22) | 90.10(0.21) |
| | EOI | **89.08(0.40)** | **88.88(0.27)** | **88.79(0.34)** | **88.01(0.29)** | **90.91(0.26)** | **91.06(0.15)** | **90.53(0.19)** | **90.45(0.39)** |

identity matrix (Lee et al., 2019a). Lastly, we explore how the computational time changes relative to the target density $d$ for a matrix with a fixed size of $n = 256$. The results of these analyses are presented in Figure 5.

By examining the Orthogonality Score, we observe that both SAO and EOI are capable of producing genuinely orthogonal matrices, while the AI method provides a cruder approximation, particularly as the network size increases.

In terms of running time with respect to the network size, we clearly observe the benefit of the $O(n)$ computation of Givens matrix multiplication in contrast to the need to compute the orthogonality loss in AI. Our method is circa 100x faster (note the logarithmic scale). Moreover, within the limit of the studied matrix sizes, it is also faster than the SAO scheme. However, we observe that SAO, being based on a structural graph construction, benefits from an overall better scaling trend, indicating that it could be a better choice for matrices with $n > 2048$.

Finally, by analyzing the plot presenting the time complexity as a function of density, we observe that EOI is especially efficient in producing extremely sparse networks – an area in which SAO performs very poorly. Once again, the scaling properties of the AI approach provide the worst running time.

The above studies confirm that our EOI scheme is an efficient way of providing exact orthogonal initializations, particularly for very high sparsity regimes and moderate matrix sizes.

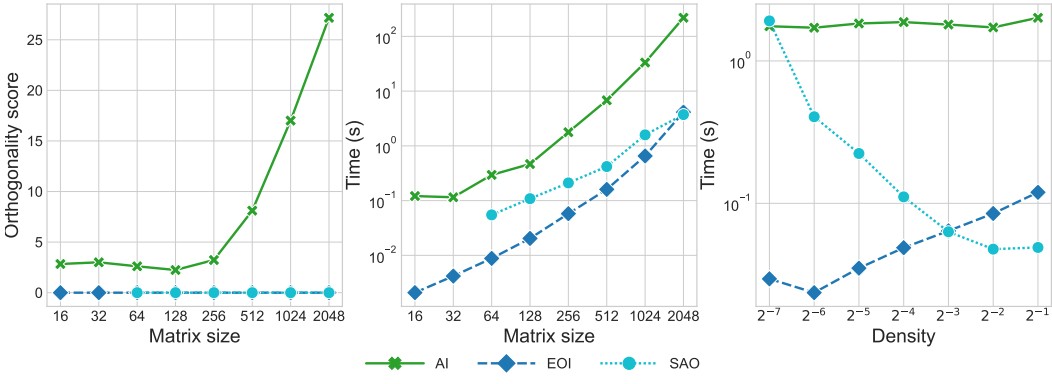

Figure 5: **Left:** The Orthogonality Score (OS) versus the matrix size for density $d = 0.0625$. **Middle:** The wall time compute versus the matrix size for density $d = 0.0625$. **Right:** The wall time compute versus density for matrix with size $n = 256$. Please note the logarithic scales in the plots.

## C   DENSITY OF GIVENS PRODUCTS

In this appendix, we examine the behavior of the process of composing random, independent Givens matrices with regard to the expected density of their product. The Givens matrix $G_n(i, j, \varphi)$ was defined in section 4.1 and can be visualized as follows:

$$
G_n(i, j, \varphi) = \begin{bmatrix}
1 & \dots & 0 & \dots & 0 & \dots & 0 \\
\vdots & & \vdots & & \vdots & & \vdots \\
0 & \dots & \cos\varphi & \dots & -\sin\varphi & \dots & 0 \\
\vdots & & \vdots & & \vdots & & \vdots \\
0 & \dots & \sin\varphi & \dots & \cos\varphi & \dots & 0 \\
\vdots & & \vdots & & \vdots & & \vdots \\
0 & \dots & 0 & \dots & 0 & \dots & 1
\end{bmatrix}, \tag{4}
$$

In our setting, we assume that $(G^{(t)})_{t \in \mathbb{N}}$ is a sequence of random, independent Givens matrices of size $n$. We define a sequence $(A^{(t)})_{t \in \mathbb{N}}$ of random matrices:

$$
A^{(0)} = I_n
$$
$$
A^{(t+1)} = A^{(t)} \cdot G^{(t)}
$$

**Theorem 1.** *Define $p(t, k)$ as the probability that the top row of $A^{(t)}$ will have exactly $k$ non-zero elements. The following recurrence relation holds:*

$$
p(t+1, k+1) = p(t, k+1) \cdot \frac{\binom{k+1}{2} + \binom{n-k-1}{2}}{\binom{n}{2}} + p(t, k) \cdot \frac{k \cdot (n-k)}{\binom{n}{2}} \tag{5}
$$

*with the base condition $p(0, 1) = 1$ and $p(0, k) = 0$ for $k \neq 1$.*

*Proof.* The base condition is trivially true since $A^{(0)}$ is the identity matrix. To prove the recurrence relation, we examine the behavior of the top row of $A^{(t)}$ when we multiply the matrix by $G^{(t)}$. Let's assume that $a$ is a row vector of width $n$, constituting the top row of matrix $A^{(t)}$, and that $G^{(t)} = G_n(i, j, \varphi)$. If $l \notin \{i, j\}$, then we can observe that $a_l$ will not change as a result of the multiplication:

$$
\left( a \cdot G^{(t)} \right)_l = a_l
$$

Otherwise, we can write:

$$
\left( a \cdot G^{(t)} \right)_i = a_i \cos\varphi + a_j \sin\varphi
$$
$$
\left( a \cdot G^{(t)} \right)_j = -a_i \sin\varphi + a_j \cos\varphi
$$

If both $a_i$ and $a_j$ are zero, then $(a \cdot G^{(t)})_i$ and $(a \cdot G^{(t)})_j$ will be zero. Otherwise, these values depend on the value of $\varphi$. If both $a_i$ and $a_j$ are non-zero, then $(a \cdot G^{(t)})_i$ and $(a \cdot G^{(t)})_j$ will be non-zero with probability 1. Finally, if exactly one of $a_i$ or $a_j$ is zero, then both $(a \cdot G^{(t)})_i$ and $(a \cdot G^{(t)})_j$ will be non-zero with probability 1. Thus, the number of non-zeros in vector $a$ can either remain the same or increase by one after multiplication by $G^{(t)}$. The probability that it remains the same is equal to the probability that $i$ and $j$ are such that $a_i$ and $a_j$ are both non-zero or both zero. On the other hand, the probability that it increases by one is equal to the probability that exactly one of $a_i$ or $a_j$ is non-zero. By calculating these probabilities and using conditional probabilities to express $p(t+1, k+1)$ in terms of $p(t, k+1)$ and $p(t, k)$, we arrive at equation 5. $\square$

**Theorem 2.** *The expected density of $A^{(t)}$ can be expressed by the following formula:*

$$
\mathbb{E}[\text{dens}(A^{(t)})] = \frac{1}{n} \cdot \sum_{k=1}^{n} k \cdot p(t, k) \tag{6}
$$

*Proof.* In Theorem 1, the values $p(t, k)$ pertain to the top row of $A^{(t)}$. However, due to symmetry, they can be applied to any row of $A^{(t)}$. In fact, the basis of Formula 5 remains the same for any row, and for any $t$, multiplication by $G^{(t)}$ independently affects every row of $A^{(t)}$, regardless of the spatial distribution of non-zeros in the row (only the number of them is relevant). Consequently, the expected density of the entire matrix $A^{(t)}$ should be equal to the expected density of any of its rows. Below, we provide a straightforward derivation of this fact, with $a$ representing the top row of $A^{(t)}$.

$$\mathbb{E}[\text{dens}(A^{(t)})] = \mathbb{E}\left[\frac{\#\{(i,j) : A_{ij}^{(t)} \neq 0\}}{n^2}\right] = \sum_{i=1}^{n} \mathbb{E}\left[\frac{\#\{j : A_{ij}^{(t)} \neq 0\}}{n^2}\right] =$$

$$\frac{1}{n} \cdot \mathbb{E}\left[\#\{j : A_{1j}^{(t)} \neq 0\}\right] = \mathbb{E}[\text{dens}(a)] = \frac{1}{n} \cdot \sum_{k=1}^{n} k \cdot p(t, k)$$

As we can see, by expressing $\mathbb{E}[\text{dens}(a)]$ as a weighted sum of probabilities, we arrive at 6. $\qquad\square$

Theorems 1 and 2 can be employed to calculate the expected densities of Givens products. Given the size of the matrices involved $n$ and an upper bound on the number of rotations $T$, we can compute all the values $p(t, k)$ for $t \in \{0, ..., T\}$ and $k \in \{1, ..., n\}$ in $O(T \cdot n)$ time using dynamic programming based on recurrence relation 5. Afterward, for each $t \in \{0, ..., T\}$, we can substitute the values $p(t, k)$ into equation 6 to determine the desired expected densities.

## D    ADDITIONAL LOSS PLOTS FOR TANH AND HARD TANH

In addition to the plots from Section 5.1, we also present the loss curves for the remaining two studied activation functions: the Tanh and Hard Tanh. Note that the Synflow method is not compatible with such activations, and hence we do report the results on this density distribution.

Similarly to the observations made in the main text, we observe that using orthogonal initializations helps to achieve better training dynamics. In addition, we also notice that sparse learning approaches based on weight initialization and function preservation such as SNIP and GraSP benefit from better learning curves than the random initializations.

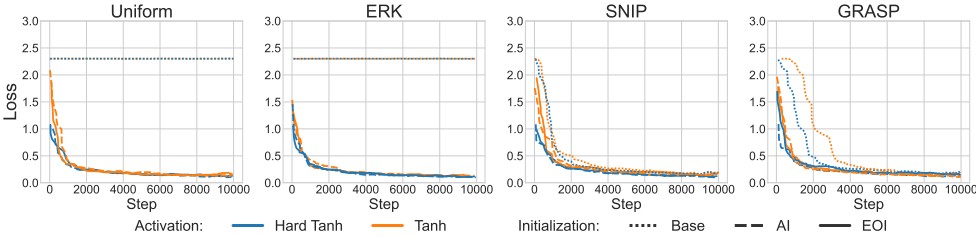

Figure 6: Training loss curve for sparsity 0.95 for Tanh and Hard Tanh activation functions.

## E    ADDITIONAL RESULTS ON CONTEMPORARY ARCHITECTURES

Apart from the main results for density 10% in Section 5.3, we also investigate the performance of the sparse initialization methods on the ResNets and VGG models for density 5%/ The results of this study are presented in Table 5.

Similarly as in the main text, we observe that the overall best performances for each model are given by the orthogonal methods. Furthermore, the EOI schemes is most often the best choice, irrespective of the used density distribution algorithm. Although our method has presented poor results when tested with SNIP, we managed to restore reasonable performance by lowering the learning rate to $10^{-2}$. In this setting, we observed results of 85.14(0.52) and 85.81(0.28) for ResNet56 and ResNet110 respectively.

Table 5: Test accuracy of various convolutional architectures on dedicated datasets, density 5%. We bold-out the best result within each density distribution method and architecture, and underline the best overall result within each architecture.

| Method | Init | ResNet32 (CIFAR10) | ResNet56 (CIFAR10) | ResNet110 (CIFAR10) | VGG-16 (CIFAR10) |
|---|---|---|---|---|---|
| ERK | Base | 87.11(0.29) | 89.11(0.28) | 90.38(0.25) | 90.06(0.20) |
| | AI | 87.25(0.16) | 88.86(0.40) | 90.21(0.27) | 90.38(0.23) |
| | EOI | **87.38(0.17)** | __89.40(0.20)__ | **90.52(0.24)** | **91.03(0.16)** |
| GraSP | Base | 86.84(0.72) | 88.89(0.16) | 90.42(0.35) | 11.81(4.04) |
| | AI | 86.78(0.41) | 89.14(0.28) | 90.35(0.27) | 11.79(4.01) |
| | EOI | **87.40(0.41)** | **89.36(0.16)** | **90.66(0.21)** | **90.03(4.36)** |
| SNIP | Base | 86.81(0.25) | 88.46(0.25) | 90.03(0.15) | 92.21(0.32) |
| | AI | 86.83(0.17) | **88.53(0.35)** | **90.18(0.32)** | **92.23(0.35)** |
| | EOI | **86.85(0.41)** | 57.09(42.99) | 25.90(35.54) | __91.97(0.16)__ |
| Synflow | Base | 86.45(0.27) | 88.32(0.18) | 89.85(0.26) | 91.36(0.09) |
| | AI | 86.27(0.30) | 88.50(0.18) | 89.96(0.33) | 91.36(0.32) |
| | EOI | **87.47(0.22)** | **89.20(0.16)** | **90.74(0.16)** | **91.86(0.23)** |
| Uniform | Base | 85.43(0.21) | 87.67(0.32) | 89.52(0.26) | 88.46(0.18) |
| | AI | 85.33(0.46) | 87.66(0.20) | 89.34(0.26) | 88.55(0.14) |
| | EOI | **86.27(0.31)** | **88.49(0.28)** | **89.99(0.09)** | **89.56(0.09)** |
| Dense | Base | 92.68(0.19) | 93.01(0.23) | 92.52(0.47) | 92.84(0.08) |
| | Orthogonal | 92.73(0.33) | 92.59(0.87) | 92.79(0.85) | 92.84(0.23) |

## F  DETAILS OF EXACT ORTHOGONAL INITIALIZATION FOR CONVOLUTIONS

In order to produce a weight tensor $\mathbf{W}^l$ of shape $c_{out} \times c_{in} \times (2k+1) \times (2k+1)$ of a convolutional layer, EOI first samples am orthogonal matrix $\mathbf{H}^l \in \mathcal{R}^{c_{out} \times c_{in}}$ generated by Algorithm 1 and sets:

$$\mathbf{W}^l_{i,j,p,q} = \begin{cases} \mathbf{H}^l_{i,j} & \text{if } p = q = k, \\ 0 & \text{otherwise.} \end{cases} \quad (7)$$

This step is adapted from *delta-orthogonal* initialization (Xiao et al., 2018), but uses a sparse matrix $\mathbf{H}^l$ instead of a dense one. In the next step, we need to establish the sparse mask $\mathbf{M}^l$. Initially, we set:

$$\mathbf{M}^l_{i,j,p,q} = \begin{cases} 1 & \text{if } \mathbf{W}^l_{i,j,p,q} \neq 0, \\ 0 & \text{otherwise.} \end{cases} \quad (8)$$

Note, however, that if $\mathbf{H}^l$ was sampled with density $d^l$, then the above equation would produce a mask with density $d^l/(2k+1)^2$. This is too low since we wanted the density of the mask (not its kernel) to be equal to $d^l$. Therefore we assume that the density of matrix $\mathbf{H}^l$ is selected to ensure that the combined density of $\mathbf{W}^l$ with embedded $\mathbf{H}^l$ does not exceed the specified target density $d^l$. To achieve the desired density, we must convert a certain number of zero entries in $\mathbf{M}^l$ into ones. We achieve it by uniformly selecting a subset of zero entries in $\mathbf{M}^l$ of the appropriate size and transforming them all to ones. Figure 8 provides a visual representation of this process.

Please note that the generated weight tensor and mask produce an orthogonal convolutional layer that aligns with the norm-preserving definition outlined in Section 3.2. The orthogonality of weights is a direct result of employing the *delta-orthogonal* initialization, which inherently satisfies the defined criteria. Notably, the mask does not prune any non-zero entries of the weights. Consequently, the orthogonality of $\mathbf{W}^l$ implies the orthogonality of $\mathbf{M}^l \odot \mathbf{W}^l$.

## G  IMPACT OF THE ROTATION ANGLE ON THE EOI ALGORITHM

One crucial variable of the distribution over Givens matrices is the angle of the rotation, denoted with $\varphi$. In algorithm 1, we sampled the angles of the Givens matrices from the uniform distribution. One might wonder how the performance of our method would change if we preferred some angles more than others. In order to verify the impact of $\varphi$ on the performance of our $EOI$, we run an experiment on the ResNet32 model with density 0.1 comparing three different distributions of Givens

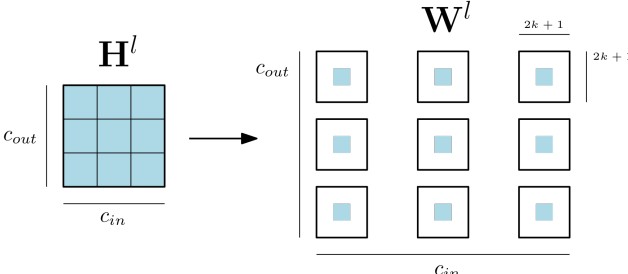

Figure 7: The embedding of an orthogonal matrix $\mathbf{H}^l$ in the convolutional tensor $\mathbf{W}^l$. Blue entries in $\mathbf{W}^l$ refer to the embedded entries of $\mathbf{H}^l$. White entries in $\mathbf{W}^l$ denote the remaining elements which are zero.

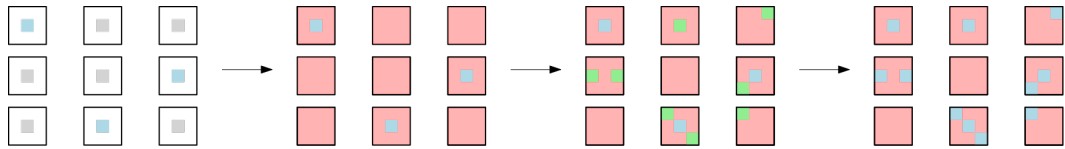

Figure 8: The process of sparse mask generation for convolutional weights. The left-most image depicts the weight matrix obtained by the embedding from Figure 7. Blue and gray entries correspond to the embedded entries of $\mathbf{H}^l$, blue representing the non-zero entries and gray representing the zeros. Next, the initial mask is applied, which prunes all zero entries (marked with red). Then, a subset of initially pruned entries is selected (marked with green) and becomes unpruned so that $\mathbf{M}^l$ matches the target density $d^l$. The final mask is depicted on the right-hand side, with blue entries being unpruned. In addition, note that entries that are not the centers of the kernels (i.e., are different from $W_{i,j,k,k}$ for any $k$) are set to zero, even if they are unpruned.

matrices: "random-angle", which corresponds to the default uniform distribution used in algorithm 1, "small-angle", which uses a fixed value of $\varphi = \pi/180$, and "large-angle", which uses a fixed $\varphi = \pi \cdot 80/180$.

We present the training loss and evaluation accuracy curves at the initial stage of the training in Figure 9. We observe that the "small-angle" method suffers from much worse convergence and higher training loss in the initial steps of the training. This indicates that such a degenerated distribution of Givens rotations isn't preferred and that larger values of $\varphi$ are more beneficial for efficient training.

## H    DESCRIPTION OF SAO

In this section, we provide a brief description of the SAO method (Esguerra et al., 2023).

The SAO algorithm is designed by leveraging a construction based on Ramanujan expander graphs. Let $G(U, V, E)$ be a $(c, d)$-regular graph, where $c$ is the degree of the input nodes, and $d$ is the degree of the output nodes. The sets $U$ and $V$ are disjoint, and any edge $(u, v) \in E$ must connect only nodes belonging to different sets. The connectivity between the sets is defined by the bi-adjacency matrix $B \in \{0, 1\}^{u \times v}$, where $u = |U|$ and $v = |V|$. It is assumed that the bi-adjacency matrix is such that the full adjacency matrix of graph $G$ is a Ramanujan expander.[3]

The SAO algorithm constructs the mask $M$ for a $m \times n$ fully-connected layer to be the transpose of the bi-adjacency matrix of a $(c, d)$-regular Ramanujan expander graph. The sets $U$ and $V$ correspond to the input and output of the layer, respectively. For $m \geq n$ and a given degree $c$, the output degree is set to $d = \frac{dm}{n}$. Next, the matrix $M$ is constructed by firstly building a $(c, 1)$-regular matrix $M_1 \in \{0, 1\}^{(n/d) \times n}$ and then concatenating its $(\frac{dm}{n} - 1)$ copies along the vertical axis (See Figure 10 for a visual comparison of examples of masked returned by SAO and EOI). Next, given $r = \frac{cn}{m}$, SAO constructs the sparse initialization matrix $S$ by generating $n/r$ sets. Each such set is

---

[3]Given a full adjacency matrix $A$ of a $d$-regular graph, the graph is said to be a Ramanujan expander if its two largest eigenvalues, $\lambda_1, \lambda_2$, satisfy $|\lambda_1 - \lambda_2| \leq 1 - \frac{\lambda_2}{d}$ and $\lambda_2 \leq 2\sqrt{(d-1)}$(Esguerra et al., 2023).

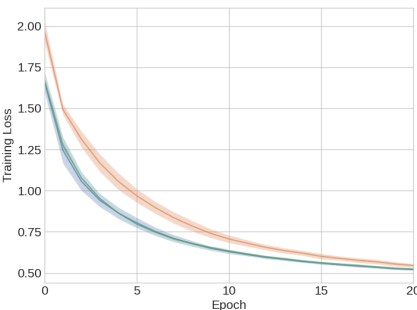 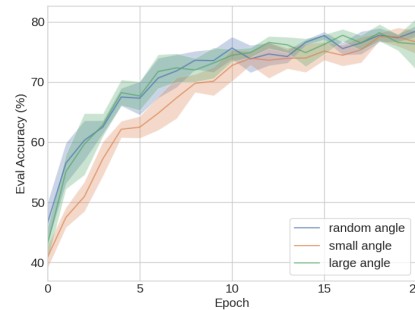

Figure 9: Mean training loss and mean evaluation accuracy curves for the first 20 epochs of training on ResNet32 with three different methods of sampling Givens matrices: "random angle", "small angle" and "large angle". Despite the differing performance in the initial stage of training, all three methods managed to finish with competitive test accuracies after 200 epochs of training - random angle: 89.58(0.09), small angle: 89.65(0.11), large angle: 89.87(0.22)

EOI                                SAO

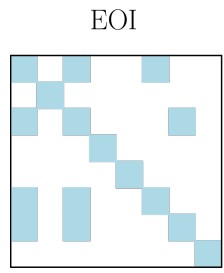 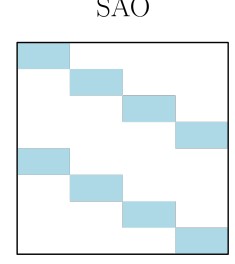

Figure 10: The visualization of example mask matrices returned by EOI (left) and SAO (right). The blue-shaded entries indicate the non-zero values.

composed of $r$ orthogonal vectors with length equal to the specified degree $c$ and corresponds to one set of orthogonal columns. The values of those vectors are then assigned to matrix $S$ accordingly to the corresponding non-zero entries of the matrix $M$ (see Figure 4 in Esguerra et al. (2023) for visualization).[4]

The density of a $(c, d)$ regular layer with weights $S \in \mathcal{R}^{m \times n}$ is given by $c/m = d/n$, where $c$ and $d$ refer to the degree of the input and output nodes, respectively. From here, it is evident that SAO introduces constraints on the possible level of sparsity, as well as the sizes of layer weights. In particular, the above construction requires that the larger dimension of the layer weights must be divisible by the specified degree $c$. Secondly, $\frac{n}{m}$ should be equal to the degree divided by some integer $r \in \mathcal{Z}$, to guarantee a degree of at least 1 in the larger dimension.

In the case of a convolutional layer, the local $k \times k$ kernel already corresponds to the $(c, d)$-sparse connection. In consequence, unlike EOI, the sparsity of mask $M$ in SAO for convolutions is controlled by pruning *entire* kernels in channels. In particular, given convolution weights of shape $c_{out} \times c_{in} \times (2k+1) \times (2k+1)$ the SAO-Delta scheme first samples a sparse orthogonal matrix $S^l \in \mathcal{R}^{c_{out} \times c_{in}}$ as described above, and then assigns $W^l_{i,j,k,k} = S_{i,j}$ and $W^l_{i,j,p,q} = 0$ if any of $p$ or $q$ is different than $k$. Each entry $S_{i,j} = 0$ will prune the *entire* kernel $W^l_{i,j}$. For more details on SAO refer to Esguerra et al. (2023).

## I    COMPARISON WITH DYNAMIC SPARSE TRAINING

---

[4]In case $n < m$, the algorithm proceeds analogously by building a $(1, d)$-regular matrix, performing the concatenation along the horizontal axis, and constructing $m/r$ sets.

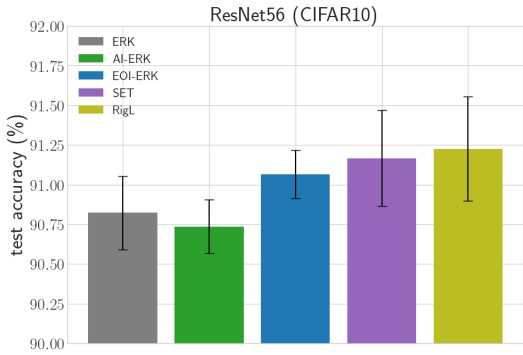

Figure 11: The CIFAR10 test accuracy obtained by the ERK-Base, ERK-AI, and ERK-EOI static pruning approaches and the SET and RigL dynamic sparse training methods. The results were computed using the ResNet56 architecture.

In addition to the experiments conducted in the main paper, we also compare the performance of our method to two commonly used dynamic sparse training approaches - SET (Mocanu et al., 2018) and RigL (Evci et al., 2020).

In dynamic sparse training, contrary to static sparse training, the initial sparse mask is allowed to change during the training. To be precise, given a masked model with density $d$, the sparse connectivity is updated after every $\Delta t$ training iterations. This update is implemented by first selecting a percentage $p\%$ of the active weights according to some pruning criterion. Next, the removed weights are replaced by new connections added accordingly to some growth criterion. For SET, the growth criterion is simply random sampling, while RigL uses the gradient norm. The total density of the model remains unchanged. The initial sparse connectivity for the DST algorithm is typically the ERK-scheme (Evci et al., 2020).

We run an experiment in which we compare the performance of SET and RigL against the ERK-Base, ERK-AI and ERK-EOI approaches on ResNet56 trained with CIFAR10. For the dynamic sparse training methods, we use an updated period of $\Delta t = 800$ and pruning percentage $p = 50\%$ which we anneal using the cosine schedule as in Evci et al. (2020). All other hyperparameters are adapted from the static sparse training regime. The results are in Figure 11. We observe that the dynamic sparse approaches indeed result in better test accuracy (as expected looking at the results of Mocanu et al. (2018); Evci et al. (2020)). However, our EOI initialization scheme, which is the best among the static sparse training methods, obtains results with low variance that are only slightly worse and still within a standard deviation from RigL and SET. At the same time, let us note that any static sparse training method can be used as an initialization point for dynamic sparse training. We consider the applicability of EOI in dynamic sparse training an interesting area for future studies.

## J  RESULTS ON IMAGENET

In this section, we consider two additional large models trained on the ImageNet dataset: the DeiT III vision transformer (Touvron et al., 2022) and ResNet50 (He et al., 2016).

**Results on ResNet.**  For the ResNet50 architecture, we use the standard hyperparameter setup with batch size 128, SGD optimizer with momentum 0.9, weight decay 0.0001, and Nesterov= `True`. We start with a learning rate equal to 0.1 and decay it after the training enters the 50% and 75% of the epochs. We use the learning rate decay of 0.1 and train the model for 90 epochs. Due to the computational intensity of this experiment, we investigate only the benefit that the EOI can bring over the ERK static sparse initialization for density=0.1. The results are presented in Figure 12. We observe that both approaches perform similarly.

**Results on DeiT.**  For the DeiT model we use the default hyperparameter setup of Touvron et al. (2022) for the "deit_small_patch16_LS" model.[5] Since DeiT is a transformer network with an attention mechanism for which the orthogonality results are not straightforward, we only apply the

---
[5]Available at `https://github.com/facebookresearch/deit`, access:14.11.2023.

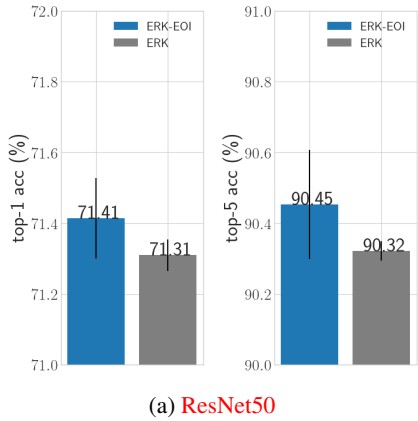 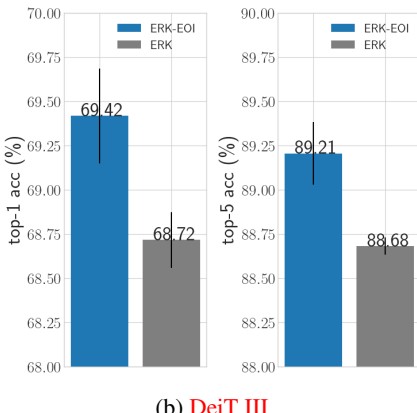

(a) ResNet50    (b) DeiT III

Figure 12: top-1 and top-5 validation accuracy on ResNet50 obtained for the ERK and ERK-EOI initializations with density 0.1.

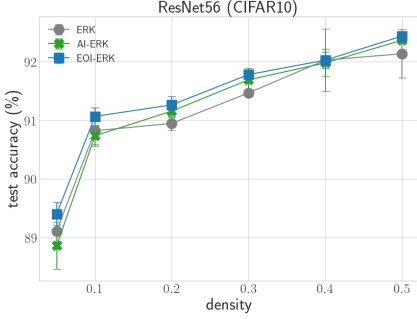

Figure 13: Test accuracy of the ResNet56 architecture trained at various sparsity levels using different sparse initialization schemes. The accuracy slowly drops between the 50% and 10% sparsity levels. Then it drops suddenly when the sparsity changes from 10% down to 3%.

sparsity to the MLP block in each encoder-building block. We again compare the base ERK with the corresponding EOI scheme on density 0.1 and we observe that the gain from exact orthogonality is clearly visible, resulting in a circa 1% increase over the ERK-base performance.

In the future, we would like to further study the possibilities of introducing sparse orthogonal weights in deep transformer networks.

## K  THE PRICE OF HIGH SPARSITY

In order to better understand how different sparsity levels affect our algorithm we conduct an experiment on the ResNet56 architecture in which we compare the performance of ERK-Base, ERK-AI, and ERK-EOI schemes for varying densities. We present the results of this setup in Figure 13.

We observe that across different densities, the best performance is given by ERK-EOI, with other methods either performing visibly worse or not holding a clear advantage. We also notice that the benefit of using EOI is the largest for the lowest densities. This is expected when compared with the results from Figure 3, in which we see that the signal propagation suffers the most in the high sparsity regime and that only EOI is able to maintain good statistics of the singular values for sparsities larger than 0.8.

