# OpenReview forum: "Sparser, Better, Deeper, Stronger: Improving Sparse Training with Exact Orthogonal Initialization"
_ICLR.cc/2024/Conference — Submitted to ICLR 2024_

### Official Review · Reviewer_4iab · 2023-10-22

**Soundness:** 4 excellent
**Presentation:** 3 good
**Contribution:** 2 fair
**Rating:** 6
**Confidence:** 4

**Summary:**

This paper introduces a new method to achieve exact (and not approximated) orthogonal sparse initialization for the weights of a (deep) neural net.
The method relays on a straightforward idea of using givens rotations, which apply an orthogonal transform on two dimensions (essentially, a 2D rotation) out of the feature dimensions. This process is repeated on random pair of dimensions, with random rotation angle, until the desired sparsity (or, density) is achieved.
The authors provide an exact formula for the expected density after a given number of rotation, which allows a precise design of the resulting initialization.

The authors provide a thorough evaluation of the method for different activation functions, under different static sparse training methods, and show compelling results when compared to *approximate* initialization.
Another comparison in done over a 1000 layer MLP with no residual connections nor normalization layers.
When trained on MNIST and CIFAR10 the proposed method achieves performance comparable to a dense network with only 12% of the weights.
Finally, the authors perform a comparison of several modern architectures on the mini-imagenet.
With the sole exception of EfficientNet, the proposed method supersedes the existing sparse approaches, and narrows the gap to dense training with only 10% of the weights.

**Strengths:**

The paper was very easy to follow and understand, the main idea is straightforward but with a clear impact.
Expected density formulation makes this method more appealing for practical usage.
The experimental section positions the method well w.r.t. existing methods.

**Weaknesses:**

While the impact is clear, it leave some questions about the price to pay for sparse networks.
For example, the reader might enjoy an analysis of the price (in performance) on the mini-imagenet for different sparsity levels.

**Questions:**

* Are some rotation angles better than others? I can't imagen that using only 1 degree Givens will perform similarly to using only 80 degree Givens
* How can one gain sense of the price (in performance) for a given sparsity level?
* One may assume that some sparsity patterns result in better performance - is that true? if so, can one guide the Givens dimensions to such a pattern

---

> ### Author Response · Authors · 2023-11-17
> **Response to Review - Part [1/2]**
>
> We express our gratitude to the Reviewer for the positive feedback regarding our work. We are delighted that the Reviewer acknowledges the clear impact of our main idea and appreciates the analysis concerning the expected density of matrices generated by our algorithm. Additionally, we were pleased to note that the Reviewer finds that our experimental section effectively positions our algorithm in comparison to existing methods. Below, we provide responses to the raised questions and suggestions:
>
> >While the impact is clear, it leave some questions about the price to pay for sparse networks. [...]
>
> Thank you for this suggestion. In order to better understand how different sparsity levels affect our algorithm, we conduct an experiment on the ResNet56 architecture in which we compare the performance of ERK-Base, ERK-AI, and ERK-EOI schemes for varying densities. We include the description and results of this experiment in Appendix K in the revised pdf. We observe that across different densities, the best performance is given by ERK-EOI, with other methods either performing visibly worse or not holding a clear advantage. We also notice that the benefit of using EOI is the largest for the lowest densities. This is expected when compared with the results from Figure 3, where we see that the signal propagation suffers the most in the high sparsity regime and that only EOI is able to maintain good statistics of the singular values for sparsities larger than 0.8.
>
> **Questions:**
>
> >Are some rotation angles better than others? I can't imagine that using only 1 degree Givens will perform similarly to using only 80 degree Givens
>
> This is a great question. In order to answer it, we run an experiment on the ResNet32 with three different distributions of Givens matrices: In the first one, we always pick 1-degree angle rotations; in the second one, we always pick 80-degree angle rotations, and in the third one, we sample the angle uniformly at random (the default method). Please see Figure 9 and the newly added Appendix G. We notice that the learning curves at the beginning of the training are indeed worse for the 1-degree rotations. This indicates that such a degenerated distribution of Givens rotations isn't preferred and that larger values of rotation angles are beneficial for efficient training.
>
> >How can one gain sense of the price (in performance) for a given sparsity level?
>
> The question of the price of performance for a given sparsity level is indeed a very interesting one. In other to assess how the performance of our algorithm is influenced by the sparsity level, we include an experiment for different densities on the ResNet56 model in Appendix K (please see also our response above). In addition, let us also note that a general intuition of how sparsity affects performance can be gained by looking into other research works in sparse training. From the empirical point of view, it has been shown that contemporary networks can be pruned up to 80% without a significant drop in performance, while for sparsity of 50% even random pruning achieves surprisingly good performance [1,2]. From the theoretical perspective, the maximal sparsity is also explored by the strong lottery ticket hypothesis, which sets limits on how large a network needs to be to have a smaller, well-performing subnetwork[3,4]. However, we would like to note that the study into empirical or theoretical limitations of sparsity was not the focus of our paper.

---

> > ### Author Response · Authors · 2023-11-17
> > **Response to Review - Part [2/2]**
> >
> > >One may assume that some sparsity patterns result in better performance - is that true? if so, can one guide the Givens dimensions to such a pattern
> >
> > The existence of well-performing, universal sparsity patterns is an intriguing question that, to the best of our knowledge, is still an open problem in sparse training. On one hand, the work of [5] studied different connectivity patterns of convolutional kernels based on graph connectivity and shows that graphs exhibiting some particular features perform better than others. On the other hand, [2] demonstrated that contemporary static sparse training approaches are invariant to re-initialization and re-shuffleing of the weights! This means that the actual mask (and, consequentially, the pattern) is not the key ingredient behind the performance of the studied in [2] methods. However, this is clearly not the case in EOI, where the reshuffling or reinitialization would break the orthogonality and hence hurt performance. Therefore, we might hope that there are some orthogonal patterns that are better than others. One could guide the EOI algorithm towards such a pattern by defining the list of axes and angles sampled for rotation in each step. One could try to define such a list by performing QR decompositions of the pattern’s matrices.
> > We leave the pursuit of such patterns for future work.
> >
> > We once again thank the Reviewer for the positive feedback. We believe we have addressed all the suggestions and questions raised by the Reviewer. It would be immensely appreciated if the Reviewer could re-assess our work and possibly reconsider raising the score. Your thoughtful reconsideration would mean a lot to us.
> >
> > **References:**
> > [1] Evci, Utku, et al. "Rigging the lottery: Making all tickets winners." International Conference on Machine Learning. PMLR, 2020.
> > [2] Frankle, Jonathan, et al. "Pruning neural networks at initialization: Why are we missing the mark?." arXiv preprint arXiv:2009.08576 (2020).
> > [3] Malach, Eran, et al. "Proving the lottery ticket hypothesis: Pruning is all you need." International Conference on Machine Learning. PMLR, 2020.
> > [4] Pensia, Ankit, et al. "Optimal lottery tickets via subset sum: Logarithmic over-parameterization is sufficient." Advances in neural information processing systems 33 (2020): 2599-2610.
> > [5] You, Jiaxuan, et al. "Graph structure of neural networks." International Conference on Machine Learning. PMLR, 2020.

---

> > > ### Comment · Reviewer_4iab · 2023-11-18
> > > **Thank you for a detailed reply**
> > >
> > > The authors have addressed all of my questions, including some very interesting results (especially Figure 9).
> > > Figure 13 show the proposed method is consistently better on several sparsity levels, even if not by much.
> > >
> > > I stand by my positive score.

---

> > > > ### Author Response · Authors · 2023-11-20
> > > >
> > > > We are immensely grateful to the Reviewer for evaluating our rebuttal and offering such valuable positive feedback throughout the review process. Given the overwhelmingly positive nature of the review and our meticulous attention to addressing all of the reviewer's inquiries, we kindly inquire whether there might be an opportunity for the Reviewer to consider adjusting the score beyond the borderline acceptance. Alternatively, if there are any remaining concerns, we are more than willing to provide any necessary information. Your support and guidance mean a lot to us, and we deeply appreciate your valuable feedback.

---

> ### Comment · Reviewer_4iab · 2023-11-22
>
> I understand the authors' desire to increase the score, and remind that the final decision is not simply based on an arithmetic mean of the scores.
> I aim to faithfully represent my opinion during the reviewer/AC discussion period, and will consider increasing the score.

---

### Official Review · Reviewer_7ET6 · 2023-10-28

**Soundness:** 3 good
**Presentation:** 3 good
**Contribution:** 3 good
**Rating:** 5
**Confidence:** 5

**Summary:**

Authors have proposed a method to achieve exact orthogonal initialization for training very deep neural models with sparsity constraints. The approach is built upon recent advancements achieved via tools from the Random Matrix theory to improve initialization and achieve training acceleration.

**Strengths:**

The proposed method is simple, mathematically grounded, and addresses an important issue of sparsity-aware training in DNNs.
The flow of the paper is nice and structured.

**Weaknesses:**

Overall, the paper is well-written, but often, critical details are missing, which might make it difficult for a new reader to understand and appreciate the ideas discussed and their connection to prior art. For instance, the majority of readers will not be able to under the SAO method. The paper should be self-contained.

The link between sparse training and sparse initialization is not clear. It is well understood that within a larger dense network, a small subnet is usually contributing the most. This contribution is different for different settings, e.g., one might want individual subnetworks to perform a task within a multi-tasking/meta-learning setting.

I request authors to support the claim that post-pruning performance is better than sparse initialization-based training. IMHO, if the parameter mask is learned or adaptive during training, the performance is usually better. The depth and architecture of the network also play a significant role in deciding up to what levels a network can be pruned, which essentially connects to the idea of effective dimension/degree of freedom given the constraint local structure imposed by the architecture.

Orthogonal initialization (sparse/non-sparse) just ensures effective signal propagation and not generalization. In the end, if the goal is effective training and achieving the best performance, how crucial is Exact Orthogonality? Unless we regularize the network, which might impact the stability, memory requirements and compute complexity.

**Questions:**

Literature:
A few missing references
- Sun et. al, Low-degree term first in ResNet, its variants and the whole neural network family
- Thukur et. al, Incremental Trainable Parameter Selection in Deep Neural Networks
- Larsson et. al, Fractalnet: Ultra-deep neural networks without residuals
- Shulgin et al, Towards a better theoretical understanding of independent subnetwork training

What are indices ijkk on page 4 in the top paragraph? A diagram of how H is embedded in a convolutional kernel of shape BxCinxCoutxHxW would help the readers.

Delta orthogonalization assumes cyclic consistency for the convolutional layer. This assumption is not discussed in the paper and might mislead the readers.

With respect to sparse static training methods, please clarify whether the pruned connections/nodes that are not updated during backward pass are used for forward pass calculation or not. In addition, it seems Masks in GraSP/Synflow are adaptive over iterations, contrary to the setting introduced by the authors.


The construction of EOI for conv kernels using random entries in the mask to achieve the desired density is not explained in detail. Is it guaranteed to be exactly orthogonal in this case, too?

Most results are empirical, and I wish authors have focused on theory to derive expressions for Dynamical Isometry in the proposed setting. Currently, the paper has half theory half numerical aspects, with both being incomplete.

Table 1: which numbers belong to MNIST and CIFAR-10?

It will be good to show the exact parameter count of each model to get a sense of how much the 10% parameters?
Then, a fair comparison would be models of the same size with different width and depth configurations.

Also, I would encourage authors to consider the full imagenet benchmark or consider a multi-task/meta-learning setting for establishing the practical benefits of the method.
In general, it looks like only the favourable experimental settings have been chosen to show the effectiveness of the approach. This is not a major issue if the main focus of paper was on theoretical aspects on the proposed initialization (which is not the case.)

---

> ### Author Response · Authors · 2023-11-17
> **Response to Review - Part [1/3]**
>
> We thank the Reviewer for the time taken to assess our work. We appreciate that the Reviewer describes our approach as simple and mathematically grounded and considers our paper well-written and structured.
>
> Before moving into answering the Reviewer's comments, we would like to clarify that the goal of our paper was to study the static sparse training setting. In such a setup, the network is already pruned (using typically unstructured methods) at initialization, and the pruning mask remains fixed throughout the training (see Section 3.3). This means that the removed connections do not take any part in the computations. In consequence, the network is sparse not only during inference (as it happens for post-pruning, including iterative pruning approaches) but also during the training. Static sparse training is also sometimes referred to as Pruning at Initialization (Frankle et al., 2020; Evci et al., 2020). We have slightly updated the paper where the terminology could have been ambiguous. Please find our responses to individual concerns below.
> >Overall, the paper is well-written [...] readers will not be able to under the SAO method. The paper should be self-contained.
>
> Thank you for pointing out the missing details. Due to the space limits, we only describe SAO briefly in the main paper, but following the Reviewer’s suggestion, we added Appendix H with a detailed description of this method and a link to this Appendix in the main paper.
> >The link between sparse training and sparse initialization is not clear.
>
> We would like to again emphasize that in this paper, we focused on the static sparse training setup (see the second paragraph of the Introduction and Section 3.3 for the definition of this setup). The goal of static sparse training is to train a sparse subnetwork by pruning a larger model before the training (see e.g. (Lee et al., 2018; Lee et al., 2019; Frankle et al., 2020; Tanaka et al., 2020; Wang et al., 2020)), and keeping the pruning mask fixed throughout the whole optimization. Therefore, every static sparse training method that computes a pruning mask M automatically defines a sparse initialization, which is simply the element-wise multiplication of the weights with their corresponding masks: $W \odot M$. We apologize, we realized we had not introduced the notion of sparse initializations clearly, which may have been a source of confusion for the Reviewer.  We have updated Section 3.3 to include the above-mentioned information (see the updated pdf, changes in red).
> >It is well understood that within a larger dense network [...] one might want individual subnetworks to perform a task within a multi-tasking/meta-learning setting.
>
> The typical setup used in the literature while comparing different static sparse training methods is a single-task problem (most often, it is a classification task (Lee et al., 2018; Lee et al., 2019; Frankle et al., 2020; Tanaka et al., 2020; Wang et al., 2020)). In order to be directly comparable with those works, as well as to be able to use the same models and datasets,  we investigated the performance of EOI in the same setups. We leave extending our results to multi-task or meta-learning setups as an interesting direction for future work
>
> >I request authors to support the claim that post-pruning performance is better than sparse initialization-based training. IMHO, if the parameter mask is learned or adaptive during training, the performance is usually better.
>
> The observation that post-pruning (i.e., pruning after training) often exceeds static sparse training (i.e., pruning at initialization) has been shown in the work of (Frankle et al., 2020). We have updated paragraph 3 of the introduction with this citation to avoid further confusion. At the same time, we would like to point out that post-training pruning and standard iterative pruning require dense training (i.e. they start with a dense model). In contrast, sparse training focuses on training architectures that are sparse (i.e. pruned) already at initialization and maintains this sparsity throughout the whole training.
> However, motivated by your comment, we also conducted an additional experiment in which we consider  an extension of static sparse training -- the dynamic sparse training. In dynamic sparse training (DST), the initial mask is allowed to change during the training, but the total density is kept fixed. We compare our EOI approach with two most common methods in DST: SET(Mocanu et al, 2020) and RigL(Evci et al., 2020). We observe that DST methods obtain better test accuracy, but our method is only slightly worse (within the bounds of the standard deviation of the DST methods) and has much lower variance.  This suggests that EOI could also be used as an initialization scheme in DST - we are eager to investigate this in the future. Please refer to Appendix I for the results and a detailed description of the experimental setting.

---

> > ### Author Response · Authors · 2023-11-17
> > **Response to Review - Part [2/3]**
> >
> > >The depth and architecture of the network also play a significant role in deciding up to what levels a network can be pruned [...]
> >
> > The question of deciding up to what levels a neural network can be pruned without a drop in performance is indeed an extremely interesting one and is studied, for example, in the work on the strong lottery ticket hypothesis (Malach et al., 2020). However, please note that the goal of our work was not to study the theoretical or empirical limits of sparsity but to propose a new sparse initialization scheme for static sparse training and compare it against other methods in this framework. Therefore, we chose the fixed levels of sparsity that had been previously used by those methods.
> >
> > >Orthogonal initialization (sparse/non-sparse) just ensures effective signal propagation and not generalization. [...]
> >
> > Please note that effective signal propagation plays a crucial role in generalization. It is well known that using an ill-defined initialization scheme will prevent the model from learning to its full capacities. Many solutions based on improving signal propagation also led to improved performance, with the ResNet architecture being probably the most prominent example. (Interestingly, it has been actually shown that the residual connection allows ResNets to naturally achieve the dynamical isometry property for any activation function (Tarnowski et al., 2019), relating it to the studies on orthogonal initializations). The disadvantages of careless initialization have also been reported in sparse training. For instance, (Evci et al., 2022) pointed out that the predominant methods for initializing sparse neural networks suffer from not  considering heterogeneous connectivity. We would also like to highlight that we do observe that using Exact Orthogonality increases the performance in contemporary networks practically in every case (Recall Figure 4 and Table 2). In addition, Table 1 shows how crucial it is to have exact isometry in order to be able to achieve better performance on extremely deep, sparse networks.
> >
> > **Questions**:
> >
> > >Literature: A few missing references [...]
> >
> > Thank you, we have added them to the related work. Please note that in (Thukur et. al) the sparsity affects only the selection of parameters for the optimizer updates. They do not perform weight pruning. The works of  (Larsson et. al), (Sun et. al) and (Shulgin et. al) all consider either dense models or models consisting of dense subnetworks with optional sparse regularization techniques. This is fundamentally different from the static sparse training approach, where every parameter in the model may be pruned in an unstructured way.
> >
> > >What are indices ijkk on page 4 in the top paragraph? A diagram of how H is embedded in a convolutional kernel of shape BxCinxCoutxHxW would help the readers.
> >
> > Earlier in the same paragraph, we note that the weights of the convolutional layer are represented with a 4-dimensional tensor W of shape (c_out, c_in, 2k+1, 2k+1). The indices i,j,k,k  mean that we select an entry i,j,p,q of that tensor such that p=q=k. We added the diagram depicting the embedding of matrix H to a new Appendix F.
> >
> > >Delta orthogonalization assumes cyclic consistency for the convolutional layer. This assumption is not discussed in the paper and might mislead the readers.
> >
> > Thank you, we mentioned this assumption in section A.2, where we say we use circular padding in the setup of the experiments for Table 1. We have now added this information to the main text in the revised paper.
> >
> > >With respect to sparse static training methods, please clarify whether the pruned connections/nodes that are not updated during backward pass are used for forward pass calculation or not.
> >
> > We only prune connections (i.e. weights), not nodes. Once a connection (i.e. weight) is pruned, it is excluded from computations both in forward and backward passes --- the weight is always zero. This is the standard approach in static sparse training - see also Section 3.3.
> >
> > >In addition, it seems Masks in GraSP/Synflow are adaptive over iterations, contrary to the setting introduced by the authors.
> >
> > Both GraSP and Synflow prune the networks at initialization. This means a mask is computed and applied by the GraSP/Synflow algorithm before training (see points 5-6 in Algorithm 1 of the GraSP paper (Wang et al., 2020), and points 1-7 of Algorithm 1 of the Synflow paper (Tanaka et al., 2020)).  However, the procedure of computing the mask can be adaptive over iterations. In particular, the key finding of Synflow is that using a data-agnostic,  iterative approach to compute the silcency scores (ipoints 2-6 in Algorithm 1 of Synflow) helps to prevent layer collapse. We use the same iterative procedure for Synflow in our experiments.

---

> > > ### Author Response · Authors · 2023-11-17
> > > **Response to Review - Part [3/3]**
> > >
> > > >The construction of EOI for conv kernels using random entries in the mask to achieve the desired density is not explained in detail [...]
> > >
> > > Thank you for pointing this out, we have added Appendix F, in which we discuss the orthogonal initialization in more detail -- see also the new Figure 8 in the Appendix. And yes, the matrix is still guaranteed to be exactly orthogonal.  Please note that we only unmask entries that are already initialized to zero, which means that we do not change any actual value of the weight matrix. Hence, the initialization remains orthogonal. Unmasking those entries simply means that they may change to non-zeros as a result of the optimization procedure (but not at initialization).
> > >
> > > >Most results are empirical, and I wish authors have focused on theory [...]
> > >
> > > Please note that the theory of dynamical isometry is quite universal, and the applicability of dynamical isometry to sparse training has already been introduced in (Lee et al., 2019). Therefore, in our work, we adopted the approach of (Lee et al., 2019), which assumes that sparsity does not disrupt the fundamental assumptions necessary to derive the dynamical isometry theory. Naturally, one could attempt to directly investigate how sparsity influences the equations fundamental to dynamic isometry defined in (Pennington et al., 2017). Please note, however, that such an endeavor requires first introducing the sparse analogy of mean field theory, then deriving formulas for Free Random Variable calculus on sparse matrices, and finally encapsulating both results to derive equations for sparse dynamical isometry. Each of these tasks requires a significant amount of work and considerably surpasses the scope of our present paper. However, we would like to emphasize that we view these tasks as exceptionally exciting and pivotal directions for future work.
> > >
> > > >Table 1: which numbers belong to MNIST and CIFAR-10?
> > >
> > > MLP uses MNIST, CNN uses CIFAR10.  We have added this information to Table 1.
> > >
> > > >It will be good to show the exact parameter count of each model to get a sense of how much the 10% parameters? [...]
> > >
> > > We have added a table in Appendix A that describes the number of parameters of each model for density 0.1.  At the same time, please note that we always use the same sparsity level whenever we compare different algorithms within one model-dataset pair.
> > >
> > > >Also, I would encourage authors to consider the full imagenet benchmark or consider a multi-task/meta-learning [...]
> > >
> > > The main goal of the paper was to introduce a new sparse initialization for (static) sparse training and compare it with other pruning-at-initialization methods. To achieve this goal, every experimental setting that we use is adapted from previous literature, either on sparse training or in orthogonal initialization. The MLP-7 from Figure 3 follows the same one used for Figure 2 in (Lee et al., 2019).  The 1000-MLPs and 1000-CNNs of Table 1 are based on the setups from (Pennington et al., 2017) and (Xiao et al., 2018), respectively. The models in Table 2 have been used in the orthogonal studies of (Lee et al., 2019), as well as in many static-sparse-training baselines (Lee et al., 2019; Tanaka et al., 2020; Wang et al., 2020). We therefore consider them to form a fair comparison setup. The adequateness of our experimental section was also recognized by Reviewer 4iab.
> > > Following the Reviewer's request, we additionally include some results on the full ImageNet dataset on the ResNet50 and DeiT III models in Appendix J. In the case of the DeIT III model we only sparsify the MLP-block, since the definition of orthogonality is not straightforward for the attention module.  In both setups, we investigate the benefit that the EOI can bring over the ERK static sparse initialization. The results are presented in Figure 12. In ResNet50, the results of ERK and ERK-EOI are quite similar, but in the DeiT III setup, EOI clearly outperforms the ERK setup. This suggests that exact sparse orthogonal initializations may be of benefit in sparsifying large transformer models.
> > >
> > > We again express our gratitude to the Reviewer for the provided feedback. We dedicated significant efforts to thoroughly address each of the concerns raised. We kindly request the Reviewer to reevaluate our work and consider a potential score adjustment.

---

### Official Review · Reviewer_s3UF · 2023-10-30

**Soundness:** 2 fair
**Presentation:** 3 good
**Contribution:** 2 fair
**Rating:** 5
**Confidence:** 3

**Summary:**

In this work, the authors propose a novel approach to sparse training, a technique aimed at training models with sparse structures from the beginning. The key element in sparse training is the sparse initialization, which determines which parts of the model are trainable through a binary mask. Existing methods often rely on predefined dense weight initialization to create these masks. However, such an approach might not efficiently harness the potential impact of these masks on the training process and optimization.

Inspired by research on dynamical isometry, the authors take an alternative route by introducing orthogonality into the sparse subnetwork. This orthogonality helps mitigate issues related to the vanishing or exploding gradient signal, ultimately making the backpropagation process more reliable.

The authors introduce their novel method called Exact Orthogonal Initialization (EOI). Unlike other existing approaches, EOI provides exact orthogonality, avoiding approximations. It also allows for the creation of layers with various densities.

**Strengths:**

The paper is well-written with a clear structure, making it easily readable for the audience.

**Weaknesses:**

However, it appears that the authors are more dedicated to highlighting the advantages of sparsity and orthogonality than proposing and demonstrating an efficient algorithm. There is a shortage of comparisons with similar algorithms in the experiments. The EOI algorithm does not seem to exhibit a significant advantage. The comparison results in Figure 3, along with the author's analysis, raise questions about whether the AI method could replace EOI. It's not clear where the innovation lies.

In Figure 5, it's unclear if the time curves represent that EOI significantly underperforms the SAO algorithm as matrix size and density increase.

In summary, the paper is well-structured and easy to read, but it lacks extensive comparisons with similar algorithms in the experiments, and the advantages of the EOI algorithm are not convincingly demonstrated. Clarity is needed in the interpretation of the results, especially in Figures 3 and 5. The meaning of bold and underlined entries in Table 3 requires clarification.

**Questions:**

P4L3: Is the k of W_{ijkk} same as the k from dimension 2k+1?
The meaning of bold and underlined entries in Table 3 is unclear, such as why some values in the 'VGG-16-GraSP-LReLU' column are bold in the middle.

---

> ### Author Response · Authors · 2023-11-17
> **Response to Review - Part [1/2]**
>
> We thank the Reviewer for the feedback. We appreciate that the Reviewer finds our paper clear and easy to follow. Below, we address the raised concerns.
>
> > However, it appears that the authors are more dedicated to highlighting the advantages of sparsity and orthogonality than proposing and demonstrating an efficient algorithm.
>
> We kindly ask the reviewer to consider that explaining the motivation behind the need of introducing a new algorithm is a crucial part of a researcher’s task. Only by understanding why we need orthogonality and what advantages it brings to sparsity can we design algorithms that effectively leverage those benefits. Note that orthogonality is commonly known to increase the stability of the training and provide better signal propagation in the network. At the same time, static sparse initializations suffer from poor gradient flow or may result in non-optimal performance (Lee et al., 2019; Evci et al., 2022). By marrying in our algorithm the good signal propagation properties stemming from orthogonality, with the reduced parameter size of static sparse training, we are able to propose a novel sparse initialization scheme that provides a boost of performance over other pruning-at-initialization methods. We do focus on demonstrating the effectiveness of our algorithm through Sections 5.2 and Sections 5.3, where EOI is able to outperform other sparse initialization approaches. It is also the most efficient one for the high sparsity regime setup (Appendix B), which lies at the core of static sparse training research.   Finally, note that since EOI produces initializations that are both sparse and orthogonal, any advantages of sparsity and orthogonality that we highlight so crucially in our work are also naturally inherited by our method.
>
> > There is a shortage of comparisons with similar algorithms in the experiments.
>
> Please note that in our work, we always include a comparison with 5 most popular static sparse training algorithms (Uniform, ERK, SNIP, GraSP, Synflow - see Section 3.3). In Table 1 and Table 2, each such method is indicated by the “<Method Name> Base” entries. Due to the findings of (Evci et al., 2022) we can also treat such methods as density-per-layer distributions for the orthogonal initialization schemes of AI and EOI (see Section 3). In consequence, for each of the 5 static sparse training methods, we can either directly use the produced by them initialization (“Base”), or treat them as a source of density-per-layer distributions for the AI and EOI algorithm, entries (“AI” and “EOI”). Moreover, for the experiments in Table 1 we also consider SAO, which has its own density distribution. Each combination of these steps yields a different variant of a sparse initialization algorithm. In summary, in Section 5.2 we evaluate in total 14 sparse algorithms, and in Section 5.3 we evaluate 15. We believe it to be a large comparison set. Let us also note that, to the best of our knowledge, AI and SAO are the only other sparse orthogonal approaches to sparse training in the literature.
>
> We recognize that our explanation of that setup might have been vague and might have been a source of confusion for the Reviewer. We have updated Sections 5.2 and 5.3 in the revised paper (changes in text in red) to include this information. In addition, we also corrected the captions of Table 1 and Table 2 to explain the meaning of the “bold” and “underscore” lines. We also encourage the Reviewer to look into our Appendix, in which we conduct additional experimental setups.

---

> > ### Author Response · Authors · 2023-11-17
> > **Response to Review - Part [2/2]**
> >
> > > The EOI algorithm does not seem to exhibit a significant advantage. The comparison results in Figure 3, along with the author's analysis, raise questions about whether the AI method could replace EOI.
> >
> > We kindly ask the Reviewer to note that the purpose of Figure 3 is only to depict (using a toy task adapted from (Evci et al., 2022)) that increasing sparsity leads to the degradation of the optimization signal. It also allows us to demonstrate that introducing orthogonality helps to mitigate this problem, motivating the validity of pursuing orthogonal sparse initializations. Figure 3 also shows the disadvantage of AI being only an approximation method, as it fails to maintain the mean and max singular values at sparsity >0.8 (corresponds to density <0.2).  Moreover, because AI is ill-specified in case of convolutions (it is not norm preserving - recall Section 3.2 and 3.3), it drastically underperforms for very deep CNN networks:   In Table 1, our EOI method achieves ~78% accuracy for the 1000-CNN experiment, while the performance of AI falls below 40%! Clearly, replacing EOI with AI would be disadvantageous.
> > We also believe that our experiments in Section 5.2 and Section 5.3 clearly depict the significant advantages in terms of increased performance when using the EOI (consider Table 1 and Figure 4). Moreover, EOI, contrary to AI, is able to achieve exact orthogonality for both FFNs and CNNs, at the same time being much more computationally efficient (Appendix B).  Moreover, unlike SAO, EOI does not introduce any limits on the sparsity level and can be used with any architecture. To explicitly point out the advantages of EOI, we included a collective summary of its benefits in the Conclusions.
> >
> > >It's not clear where the innovation lies.
> >
> > The proposed EOI initialization scheme is a new construction for sparse orthogonal initialization. It is the first orthogonality-based sparse initialization that, at the same time, provides exact orthogonality, works perfectly for convolutional layers,  supports any density-per-layer, and does not unnecessarily constrain the layer’s dimensions. EOI is also data-agnostic (i.e., it does not require access to the training data to compute the masks).  This combination allows EOI to outperform other approaches in all experiment setups, making it an impactful piece of work in the area of static sparse training. The simplicity of our method and its importance in the field of sparse training has also been appreciated by Reviewers 4iab and 7ET6.
> >
> > >In Figure 5, it's unclear if the time curves represent that EOI significantly underperforms the SAO algorithm as matrix size and density increase.
> >
> > In the case of the time vs. matrix size curve, we see that EOI is the best choice for matrices with sizes smaller than 2048. For matrix sizes larger or greater than 2048 we expect that the SAO method will give better time results, as it exhibits an overall better scaling trend. We have commented on this exact fact in the last two sentences of paragraph 3 in Appendix B. This means that EOI is the preferred solution for matrices smaller than 2048. This is a significant advantage, since in practical scenarios, larger widths are seldom employed. In the case of time vs. density, it is quite evident that EOI is especially efficient in producing extremely sparse networks (for densities smaller than $2^{-2}$). In this area, SAO performs very poorly (we commented on that in paragraph 4 of Appendix B). Note that in static sparse training, we are always interested in the high sparsity regime (with density smaller <0.2), hence EOI would be better suited in such setups (we also write about it last paragraph of Appendix B).  On top of that, please note  that SAO allows only for incremental sparsity and cannot be applied to arbitrary types of matrices, rendering it impossible to use in contemporary architectures.
> >
> > **Questions**
> >
> > > P4L3: Is the k of W_{ijkk} same as the k from dimension 2k+1?
> >
> > Yes, it is the same. Thank you for pointing that out, we clarified that in Section 4.2.
> >
> > >The meaning of bold and underlined entries in Table 3 is unclear, such as why some values in the 'VGG-16-GraSP-LReLU' column are bold in the middle.
> >
> > Thank you for noticing that such an important piece of information was missing. The bolded entries denote the best results within a fixed per-layer density distribution method and architecture. The underlined entries denote the best overall result within a fixed architecture. We have updated the captions in the revised pdf.
> > We thank the Reviewer for the feedback. We are confident that we have adequately addressed all the raised concerns. Should there be any further inquiries, we are available and eager to assist the Reviewer. Additionally, we kindly request the Reviewer to reconsider our work at their convenience.

---

> > > ### Comment · Reviewer_s3UF · 2023-11-22
> > >
> > > Thank you for your efforts in addressing my concerns during the rebuttal phase. While I appreciate your detailed responses, I must convey that I am unable to modify my opinions at this time.

---

> > > > ### Author Response · Authors · 2023-11-22
> > > >
> > > > Thank you for the response. Could the Reviewer elaborate more on the reasoning behind the decision — is the Reviewer dissatisfied with a point in our response, or perhaps are there any other concerns which the Reviewer would like us to address? We greatly appreciate your feedback.

---

### Author Response · Authors · 2023-11-17
**Joined Response**

We thank all the Reviewers for the time and effort spent to provide valuable insights and comments on our work which allowed us to significantly improve the paper. We are grateful that our method has been praised for being simple, mathematically grounded, and impactful (Reviewers 7ET6, 4iab). We appreciate that our experiments have been acknowledged to adequately position our EOI algorithm with respect to existing methods (Reviewer 4iab). We are also pleased to learn that all the Reviewers found our paper to be well-written and easy to read.  Overall, we are delighted to receive such an encouraging response from the Reviewers regarding the aforementioned aspects of our research. Below, we address specific comments raised by Reviewers and highlight the updates made to the paper.

---

> ### Author Response · Authors · 2023-11-22
>
> We would like to once again thank all the Reviewers for the feedback. We believe we have addressed all the raised concerns and improved our paper following the Reviewers’ suggestions. As the discussion period approaches its conclusion, we kindly request the Reviewers to consider possibly adjusting the score, or alternatively, bring to our attention any further inquiries. We appreciate your time and consideration.

---

### Meta-Review · Area_Chair_y9VX · 2023-12-03

**Metareview:**

This paper is about sparse training. While appreciating the novelty of the proposed technique, the reviewers raise various concerns, which have not been fully addressed by the author’s rebuttal. According to my own reading, I agree with the reviewers that the experimental validations are quite weak. Only two simple datasets, MINIST and CIFAR10, are considered. The experimental results are poorly presented, making hard for the readers to see the superiority of the proposed method. Overall, I think the paper contains certain merits but the current version cannot meet the standard of this conference. I would recommend rejecting the paper.

**Justification For Why Not Higher Score:**

While appreciating the novelty of the proposed technique, the reviewers raise various concerns, which have not been fully addressed by the author’s rebuttal. According to my own reading, I agree with the reviewers that the experimental validations are quite weak. Only two simple datasets, MINIST and CIFAR10, are considered. The experimental results are poorly presented, making hard for the readers to see the superiority of the proposed method.

**Justification For Why Not Lower Score:**

N/A

---

### Decision · Program_Chairs · 2024-01-16

Reject